

# Estimating $N_2O_5$ uptake coefficients using ambient measurements of $NO_3$, $N_2O_5$, $ClNO_2$ and particle-phase nitrate.

G. J. Phillips[1], J. Thieser[1], M. J. Tang[1], N. Sobanski[1], G. Schuster[1], J. Fachinger[2], F. Drewnick[2], S. Borrmann[2], H. Bingemer[3], J. Lelieveld[1], and J. N. Crowley[1]

[1]Department of Atmospheric Chemistry, Max Planck Institute for Chemistry, Germany
[2]Particle Chemistry Department, Max Planck Institute for Chemistry, Germany
[3]Institute for Atmospheric and Environmental Sciences, Goethe University, Frankfurt, Germany

*Correspondence to*: John Crowley (john.crowley@mpic.de)

**Abstract.** We present an estimation of the uptake coefficient ($\gamma$) and yield of nitryl chloride ($ClNO_2$) ($f$) for the
heterogeneous processing of dinitrogen pentoxide ($N_2O_5$) using simultaneous measurements of particle and trace gas composition at a semi-rural, non-coastal, mountain site in the summer of 2011. The yield of $ClNO_2$ varied between ($0.035 \pm 0.027$) and ($1.38 \pm 0.60$) with a campaign average of ($0.49 \pm 0.35$). The large variability in $f$ reflects the highly variable chloride content of particles at the site. Uptake coefficients were also highly variable with minimum, maximum and average $\gamma$ values of 0.004, 0.11 and $0.028 \pm 0.029$, respectively, with no significant correlation with particle composition, but a weak
dependence on relative humidity. The uptake coefficients obtained are compared to existing parameterisations based on laboratory datasets and with other values obtained by analysis of field data.

## 1 Introduction

The reaction of $N_2O_5$ with atmospheric aerosol represents an important nocturnal control on the atmospheric lifetime of NO$x$ and can impact the production of atmospheric oxidants, such as ozone and hydroxyl radicals (Dentener and Crutzen, 1993;
Riemer et al., 2003; Brown et al., 2006; Macintyre and Evans, 2010). The reaction, in addition to loss of NO$x$, results in the formation of particulate nitrate and can result in the release of $ClNO_2$ in both marine and continental environments (Osthoff et al., 2008; Kercher et al., 2009; Thornton et al., 2010; Mielke et al., 2011; Phillips et al., 2012; Riedel et al., 2012a; Bannan et al., 2015).

Neglecting gas-phase diffusive effects, which are insignificant for transport of $N_2O_5$ to the sub-micron diameter particles
dealt with in this study, the rate of loss of $N_2O_5$ on a particle surface can be described by the following expression:

$$\frac{d[N_2O_5]}{dt} = -0.25\overline{c}\gamma A[N_2O_5] \tag{1}$$

where $\overline{c}$ is the mean molecule speed of $N_2O_5$, $\gamma$ is the uptake coefficient, and $A$ is the aerosol surface area density (i.e. the particle surface area per volume of air). This expression is the basis of the numerous laboratory studies designed to derive $\gamma$





for tropospheric and stratospheric particles and its dependence on environmental variables such as temperature and relative humidity (Ammann et al., 2013).

The reaction of $N_2O_5$ with aqueous particles is complex. Following bulk accommodation, the uptake proceeds via disproportionation of $N_2O_5$ in the aqueous phase (Mozurkewich and Calvert, 1988) to form $HNO_3$ (and $NO_3^-$):

$$N_2O_5 \text{ (aq)} + H_2O \text{ (l)} \qquad \leftrightarrow H_2NO_3^+ \text{ (aq)} + NO_3^- \text{ (aq)} \qquad \text{(R1, R-1)}$$

$$H_2NO_3^+ \text{ (aq)} + H_2O \text{ (l)} \qquad \leftrightarrow HNO_3 \text{ (aq)} + H_3O^+ \text{ (aq)} \qquad \text{(R2, R-2)}$$

$$HNO_3 \text{ (aq)} + H_2O \text{ (l)} \qquad \leftrightarrow NO_3^- \text{ (aq)} + H_3O^+ \text{ (aq)} \qquad \text{(R3, R-3)}$$

In the presence of chloride, $ClNO_2$ can also be formed (Finlayson-Pitts et al., 1989; George et al., 1994; Behnke et al., 1997).

$$H_2NO_3^+ \text{ (aq)} + Cl^- \text{ (aq)} \qquad \to ClNO_2 + H_2O \text{ (l)} \qquad \text{(R4)}$$

The relative rate of formation of the $NO_3^-$ (or $HNO_3$) and $ClNO_2$ products of R2 and R4 depends on the fate of $H_2NO_3^+$ (hydrated nitronium ion) and thus on the concentration of chloride available (Behnke et al., 1997; Schweitzer et al., 1998; Thornton and Abbatt, 2005; Bertram and Thornton, 2009; Roberts et al., 2009) and on the rate coefficients for its aqueous phase reaction with $Cl^-$ ($k_4$) or $H_2O$ ($k_2$).

We define the branching ratio to $ClNO_2$ formation as $f$ so that the net yield of $NO_3^-$ and $ClNO_2$ formed (per $N_2O_5$ taken up)

can be written as:

$$N_2O_5 + (H_2O \text{ or } Cl^-) \qquad \to (2-f) NO_3^- + f ClNO_2 \qquad \text{(R5)}$$

$f$ can be calculated from knowledge of the relative chloride and water content of a particle and the rate coefficients $k_2$ and $k_4$ :

$$f = \frac{k_4[Cl^-]}{k_4[Cl^-] + k_2[H_2O]} \qquad (2)$$

A value of $k_4 / k_2$ of 450 is presently recommended (Ammann et al., 2013), but may be modified for systems with a

significant concentration of aromatics which can also react with hydrated nitronium ions (Ryder et al., 2015).

Nitrate formed in the uptake of $N_2O_5$ can partition to the gas phase as nitric acid, or remain in the particle depending on the pH, available ammonia/ammonium and temperature. Apart from highly acidic aerosol (Roberts et al., 2009) $ClNO_2$ is only weakly soluble / reactive and degasses completely from the particle. In the absence of appreciable chemical losses at night, $ClNO_2$ remains in the atmosphere until dawn when it is removed by photolysis over a period of a few hours (Ghosh et al.,

25    2012).

$$ClNO_2 + h\nu \ (\lambda < 852 \text{ nm}) \to Cl + NO_2 \qquad \text{(R6)}$$

The photochemical destruction of $ClNO_2$ recycles $NOx$ and activates $Cl$ atoms for possible oxidation of VOCs, which may lead (via peroxy radical formation) to enhanced NO to $NO_2$ oxidation and ozone production (Simon et al., 2009; Sarwar et al., 2012). The chemical processes involved in the nighttime formation of $N_2O_5$, its heterogeneous processing and the

subsequent daytime photochemical reactions involving $Cl$ atoms (and their impact on VOC oxidation and ozone formation) are illustrated in Fig. 1.

Laboratory studies (using synthetic surrogates for atmospheric aerosol) indicate that $N_2O_5$ uptake coefficients on aqueous, tropospheric aerosol are large (e.g. $1\text{-}3 \times 10^{-2}$ for ammonium sulphate) and show a complex dependence on environmental



variables such as temperature and relative humidity and also on the nitrate and chloride content of the particle. A detailed summary of the results of the laboratory studies is given by Amman et al. (2013).

Briefly, the presence of particulate nitrate has been seen to reduce $\gamma$, which is understood in terms of enhancing the rate of (R-1) compared to R2 and R4 (Wahner et al., 1998a; Wahner et al., 1998b; Mentel et al., 1999; Hallquist et al., 2003; Bertram and Thornton, 2009). In contrast, the presence of chloride increases $\gamma$ through the competitive removal of $H_2NO_3^+$ in R4 (Bertram and Thornton, 2009). Laboratory experiments have also documented the reduction of $\gamma$ caused by particle phase organics, including organic coatings, which lower the water activity and the hydrolysis rate of $N_2O_5$ and possibly the rate of accommodation of $N_2O_5$ at the gas-particle interface (Anttila et al., 2006; Badger et al., 2006; McNeill et al., 2006; Park et al., 2007; Cosman and Bertram, 2008; Cosman et al., 2008; Gaston et al., 2014). The uptake coefficient to dry particles is reduced for the same potential reasons (Hu and Abbatt, 1997; Thornton et al., 2003).

The loss of $N_2O_5$ to ambient aerosol samples (as opposed to chemically "simpler" aerosol surrogates prepared in the laboratory) was investigated by Bertram and colleagues (Bertram et al., 2009a; Bertram et al., 2009b; Riedel et al., 2012b) who generated gas-phase $N_2O_5$ and monitored its loss via uptake to atmospheric aerosol sampled into a flow-reactor. The results indicated that the uptake coefficient was highly variable, sometimes approaching a factor of 10 lower than derived from laboratory studies using synthetic aerosol and strongly dependent on aerosol composition, especially the water / organic content. Other, less direct measurements of $N_2O_5$ reactivity on ambient particles (discussed in detail later), indicate high variability and occasional low values that are inconsistent with laboratory investigations on pure samples.

In this paper we address the need for measurements of the $N_2O_5$ uptake coefficient to particles in the real atmospheric environment. We present simultaneous measurements of particle composition, $N_2O_5$ and $ClNO_2$ to derive both the uptake coefficient ($\gamma$) and the efficiency ($f$) of $ClNO_2$ production in number of different air masses and discuss the limitations and validity of this and similar approaches.

## 2 Methods

### 2.1 PARADE field site

The PARADE measurement intensive (Particles And RADicals: diel observations of the impact of urban and biogenic Emissions) took place between mid-August to mid-September 2011 at the Taunus Observatory (TO), Kleiner Feldberg, Germany. The observatory is operated by the Institute for Atmospheric and Environmental Science of the Goethe University, Frankfurt. The Kleiner Feldberg (825 m above sea-level) sits within the Taunus range approximately 30 km NW of Frankfurt am Main, Germany. The site, which may be described as rural with anthropogenic impact from local industrial / population centres, is described in more detail by, Wobrock et al. (1994), Handisides (2001), Crowley et al. (2010) and Sobanski et al. (2016). It is $\approx$ 400 km distant from the nearest (North Sea) coastline.



## 2.2 Instrumentation

$N_2O_5$ and $NO_3$ were measured using cavity-ring-down spectroscopy (CRDS) with instruments described by Schuster et al. (2009) and Crowley et al. (2010). $NO_3$ was measured directly in a cavity at ambient temperature whereas the sum $N_2O_5$ + $NO_3$ was measured in a separate cavity at ~100 °C after thermal decomposition of $N_2O_5$ to $NO_3$ in a heated section of the inlet, also at 100 °C. Following corrections for the transmission of $NO_3$ and $N_2O_5$ though the inlets, filter and cavities, the difference signal is used to calculated $N_2O_5$ mixing ratios.

The CRDS was situated on a platform on the roof of the TO-laboratory at a height of 10 meters from the ground. Air was drawn through a 1 m length of ½" (OD) Teflon (perfluroalkoxy, PFA) tubing at 50 (std) Lmin$^{-1}$ (SLM) and sampled (18 SLM) from the centre of the flow via ¼" PFA tubing and a Teflon filter (exchanged hourly using an automatic filter changer) into the two cavities of the CRDS. This setup keeps inlet residence times short (~0.1 s) and also reduces sampling of coarse particles and droplets. The total uncertainty associated with the $N_2O_5$ measurements was 15 % with the largest contribution the uncertainty in the $NO_3$ cross section and $NO_3$ losses.

The non-refractory composition of particulate matter with an aerodynamic diameter less than 1 µm (PM1) was measured with an Aerodyne HR-ToF aerosol mass spectrometer (AMS)(Jayne et al., 2000; DeCarlo et al., 2006). The AMS was deployed on board the MPIC mobile laboratory (MoLa)(Drewnick et al., 2012) located approximately 15 meters from the measurement station with air sampled from ~ 7 m above the ground via wide bore metal tubing at a flow rate of 90 (STD) L min$^{-1}$. Total organic, nitrate, sulphate and chloride in non-refractory particles of < 1 µm diameter (NR- PM1) are reported here. As the AMS detects marine chloride (i.e. refractory chloride) with one to two orders of magnitude lower sensitivity than non-refractory chloride (Zorn et al., 2008; Ovadnevaite et al., 2012; Schmale et al., 2013; Drewnick et al., 2015) it is reasonable to assume that the majority of the Cl$^-$ reported by the AMS is due to $NH_4Cl$ arising from uptake of gas-phase HCl which may have marine or anthropogenic origin. The AMS was calibrated using the measurement of standard ammonium nitrate particles of known particle size via a DMA system.

Particle size spectra were measured using an optical particle counter (OPC, Model 1.109,Grimm), an aerodynamic particle sizer (APS, Model 3321, TSI) and a fast mobility particle size spectrometer (FMPS, Model 3091, TSI). All particles were sampled at ambient RH so corrections for hygroscopic particle growth were not necessary. The uncertainties associated with parameters required for calculation of the uptake coefficient are 25 % for the nitrate measurement and 30 % for the particle diameter measurement (Wiedensohler et al., 2012). Note that an uncertainty in the particle diameter of ~30 % implies an uncertainty in the particle surface area of ~ 70 %. This uncertainty applies not only to this study, but also to all previous laboratory and field studies that use similar instrumentation and thus a certain cancelling of errors occurs when comparing to other datasets.

$NO_2$ was measured during PARADE using a variety of techniques which all showed good agreement. The data used in this analysis were obtained using the MPIC $NO_2$ TD-CRDS system (Thieser et al., 2016). The instrument is a two-cavity system which also measured total peroxy nitrates and total alkyl nitrates ($\Sigma$PNs and $\Sigma$ANs). The atmosphere was sampled by



drawing air at ~20 SLM through a 3/8" outer-diameter PFA tube 8 m in length and sub sampling approximately 4 SLM into the cavity. The main inlet was shared by the iodide CIMS system which was used to measure $ClNO_2$, speciated peroxy nitrates and peroxy acids (Phillips et al., 2012; Phillips et al., 2013b).

The $ClNO_2$ dataset and method was described by Phillips et al. (2012) and follows prior measurements of $ClNO_2$ using the same technique (Osthoff et al., 2008; Thornton et al., 2010; Mielke et al., 2011). The CIMS instrument was constructed by THS Instruments, Georgia, USA and is based on the CIMS technique described by Slusher et al. (2004) and Zheng et al. (2011). No indication of the production of $ClNO_2$ on the inlet walls was observed. $ClNO_2$ was monitored at $I^{37}Cl^-$ (m/z = 163.9, with a LOD ($2\sigma$) of 12 pptv) for the entire period, however there are clear indications (i.e. non-zero daytime concentrations) that the signal at this mass is not solely due to $ClNO_2$ and consequently $IClNO_2^-$ (m/z = 207.9) was monitored with a LOD ($2\sigma$) of 3 pptv, from the 02.09.2011. The impurity at $m/z$ = 163.9 contains one chlorine atom ($m/z$ = 161.9 was also observed at the correct isotope ratio) and has a diel cycle that strongly resembles that of HCl, which was measured using ion-chromatography during a subsequent campaign (Phillips et al., 2013a) at the same site and time of year. The mechanism of HCl detection at $m/z$ = 161.9 and $m/z$ = 163.9 remains unclear as the reaction $I^- + HCl \rightarrow ICl^- + H$ is endothermic by at least 200 kJ/mol.

The instrument sensitivity to $ClNO_2$ was determined by the measurement of a $ClNO_2$ standard synthesized by passing $Cl_2$ over a mixture of $NaNO_2$ and $NaCl$ crystals in a flow of humidified $N_2$. The concentration of the standard was determined using thermal-dissociation cavity ring-down absorption spectroscopy (TD-CRDS) using the instrument described by Thieser et al. (2016). A zero measurement, using a bypass with 25 cm of metal wool heated to 473 K, was made once an hour and the accuracy of the $ClNO_2$ measurement is 25 % (Phillips et al., 2012).

Meteorological data were obtained from the public data depository of the Hessisches Landesamt für Umwelt und Geologie (HLUG) monitoring station situated at the peak of the Kleiner Feldberg approximately 10 meters from the main sample inlet location. Data is available from http://www.hlug.de/start/luft/luftmessnetz.html. Back trajectories were calculated using HYSPLIT (Draxler and Rolph, 2011; Stein et al., 2015).

## 3 Meteorological / chemical conditions during PARADE

The time series of a selection of particle and trace-gas concentrations measured during PARADE is shown in Figs. 2 and 3 along with temperature and relative humidity.

### 3.1 Particle characteristics

Fig. 2 indicates that the aerosol surface area ($A$) available for uptake of $N_2O_5$ was highly variable and was predominantly (> 75 % on average) associated with particles less than 550 nm in diameter (FMPS data). Relative humidity, an environmental parameter which influences the water content of the particles and is thus expected to influence the uptake of $N_2O_5$ significantly, varied between 25 and 100 %. On average, the sub-micron non-refractory aerosol was (by mass) 55 % organic,



26 % sulphate with nitrate and ammonium both 9.5 %, which may be considered typical for an anthropogenically influenced, rural region. Particulate sulphate and organic content were correlated, with the particulate sulphate to organic mass ratio, a parameter that potentially impacts on the $N_2O_5$ uptake to particles (Bertram et al., 2009b), varying between ~0.1 and 2.4 with a mean value of ~0.6. The campaign averaged nitrate particle mass concentration was 0.56 µg m$^{-3}$ during the day and 0.89

µg m$^{-3}$ during the night at this site. The larger nighttime values reflect high rates of nitrate production from $N_2O_5$ uptake at nighttime rather than temperature dependent partitioning of $HNO_3$ / $NH_3$ / ammonium nitrate (Phillips et al., 2013a), which we discuss below.

**3.2 NO$x$, O$_3$ and N$_2$O$_5$ formation**

As reported previously for this site (Crowley et al., 2010; Phillips et al., 2012; Sobanski et al., 2016) local emissions result in

high variability in NO$x$ with $NO_2$ usually between ~0.5 and 10 ppbv but with excursions up to ~ 20 ppbv. Daytime maxima for NO were between ~ 0.5 and 2 ppb and ozone levels were between 20 and 70 ppbv. The high variability in $NO_2$ and $O_3$ result in a highly variable $NO_3$ production term between <0.05 and >0.5 pptv s$^{-1}$ (Sobanski et al., 2016). The $NO_3$ and $N_2O_5$ lifetimes were also highly variable and on some nights (notably 20-21.08, 30-31.08, 31.08-01.09, 01-02.09) on which extended $NO_3$ (and $N_2O_5$) lifetimes were observed we have compelling evidence that the inlets were sampling from a

relatively low altitude residual layer (Sobanski et al., 2016).

**3.3 Meteorology, ClNO$_2$ and particulate nitrate**

The PARADE measurement period began on 15.08 and ran until 17.09. As described by Phillips et al. (Phillips et al., 2012), this period may be separated into three meteorologically distinct parts.

*Period 1 (17.08 – 26.08):* The period beginning on the 17.08 and ending on 26.08 was changeable with calculated air mass

24 hour back-trajectories suggesting the air was largely continental in origin, arriving from the west-to-south wind sector. The exceptions to the continental origin occurred up to sunrise on the 17.08 and on the 19-20.08 with air arriving at the site with relatively high humidity from the NW with 48-hour back-trajectories indicating an approximate UK/English channel origin. On the nights of the 17-18.08 and 19-20.08 concurrent increases in ClNO$_2$ and NR PM1 NO$_3^-$ were observed. There is little indication of nocturnal production of ClNO$_2$ on the remaining nights of this initial period of the measurements. In

fact, on the night of the 20-21.08 high concentrations of $N_2O_5$ were measured, $[N_2O_5]_{max} \approx 800$ pptv, during a period of low RH of 25 % (lowest observed during the measurement period). No concurrent increase in the concentration of submicron particulate nitrate or ClNO$_2$ was measured.

*Period 2 (26.08 – 29.08):* On the evening of 26.08 a front passed across the measurement site associated with heavy cloud cover and rain. Before the passage of the front, large, variable concentrations of NO$x$ were measured coinciding with an

increase in PM1 particle mass. The rain was sustained following the passing of the front and a RH of 100% was measured into the morning of 28.08. After the front, concentrations of fine particles, as measured by the FMPS and AMS, and NO$x$



were relatively low and remained suppressed until 29.08, with the exception of NR PM1 Cl$^-$ which peaked on the nights of 27-28.08 and 28-29.08 possibly due to the marine influence of the post-frontal air and the increased supply of chloride from marine particles. Following 29.08 large concentrations of ClNO$_2$ were observed on a series of nights up to and including 3.09. Back-trajectories calculated with the HYSPLIT model suggest that the westerly flow following the front gradually weakened and a surface high pressure set in leading to high concentrations of NO$x$ and O$_3$ probably due to the influence of the near conurbation of Frankfurt-Wiesbaden-Mainz.

*Period 3 (29.08- 09.09):* The remainder of the measurement period was influenced by a mainly westerly flow and the observatory was frequently shrouded with cloud. Nocturnal production of modest concentrations of ClNO$_2$ occurred between 6.09 and 9.09 without a concomitant increase in NR PM1 NO$_3^-$. Each of these overnight periods was impacted by cloud at the measurement site with the exception of the 6.09 where the RH was nevertheless above 90%. Measurements of particle composition during PARADE were only available at the submicron size; it may be possible that during misty and cloudy conditions the nitrate formation occurred on surfaces which were not measured, i.e. larger mist or fog droplets or, in some cases nitrate formed was scavenged within the clouds. Photolysis frequencies were attenuated by the cloud cover and rain at the site during this period, allowing measurable quantities of ClNO$_2$ to persist well past noon, see for example 8.09 to 9.09. N$_2$O$_5$ was not measured beyond 9.9; however, the little data available during this period suggest that relative productivity with respect to ClNO$_2$ was high.

## 4 Estimation of $f$ and $\gamma$ for N$_2$O$_5$ uptake to ambient aerosol

Apart from the measurement of the loss rate of synthetic N$_2$O$_5$ samples to ambient aerosol as carried out by Bertram and co-workers (see section 1), there are different methods by which ambient measurements can be analysed to derive an uptake coefficient ($\gamma$) for N$_2$O$_5$ interaction with atmospheric particles, which depend on the data available. These include 1) the analysis of product formation (gas and/or particle phase) resulting from the uptake of N$_2$O$_5$ (Mielke et al., 2013), 2) analysis of the steady-state N$_2$O$_5$ (or NO$_3$) lifetime (Brown et al., 2006; Brown et al., 2009; Brown et al., 2016a) and 3) box modelling of N$_2$O$_5$ chemistry with various observational constraints (Wagner et al., 2013). All three methods have their own strengths and weaknesses and all contain assumptions that are evidently not applicable in all cases.

We first discuss the methods which we have applied (1 and 2) to derive $f$ and $\gamma$ during the PARADE measurement campaign and compare the results to similar analyses in the literature.

### 4.1 Method 1: Using product formation rates to derive $f$ and $\gamma$

For the PARADE campaign, periods of data were identified where clear correlations between particulate nitrate NO$_3$ and ClNO$_2$ were observed, as illustrated, for example, in Fig. 4. On these nights, there is no particulate nitrate formation without





concurrent $ClNO_2$ formation and the covariance of $NO_3^-$ and $ClNO_2$ is taken as evidence that, to a good approximation, both $ClNO_2$ and particulate nitrate are produced only by the uptake of $N_2O_5$.

From Eq.1 and the definition of the branching ratio (R5), the rate of production of $ClNO_2$ ($pClNO_2$) from the reaction of $N_2O_5$ with aerosol (surface area $A$) can be written as:

$$pClNO_2 = \frac{d[ClNO_2]}{dt} = f(0.25\gamma\bar{c}A[N_2O_5]) \qquad (3)$$

The rate of production of particulate nitrate ($pNO_3^-$) is:

$$pNO_3^- = \frac{d[NO_3^-]}{dt} = 2(1-f)(0.25\gamma\bar{c}A[N_2O_5]) + f(0.25\gamma\bar{c}A[N_2O_5]) \qquad (4)$$

Combining and rearranging Eq.3 and Eq.4 we get:

$$f = 2\left(\frac{pNO_3^-}{pClNO_2} + 1\right)^{-1} \qquad (5)$$

and

$$\gamma = \frac{2(pClNO_2 + pNO_3^-)}{\bar{c}A[N_2O_5]} \qquad (6)$$

Note that the fractional formation of $ClNO_2$ ($f$) is the amount of $ClNO_2$ formed per $N_2O_5$ taken up to a particle and should not be confused with the yields of $ClNO_2$ per $NO_3$ formed that have occasionally been reported (Osthoff et al., 2008; Mielke et al., 2013) and which represent lower limits to $f$ as $NO_3$ is not necessarily converted stoichiometrically to $N_2O_5$ in the atmosphere.

Eq.5 illustrates that measurement of the production rates of $ClNO_2$ and particulate nitrate is sufficient to derive the fractional yield of $ClNO_2$ ($f$) and that, by also measuring the concentrations of $N_2O_5$ and aerosol surface area ($A$), we can derive the uptake coefficient, $\gamma$ (Eq.6). Note that without measurement of both particulate nitrate and $ClNO_2$ production, it is not possible to derive both the yield of $ClNO_2$ and the uptake coefficient. By analysing the $ClNO_2$ product alone, Eq. 3 can be used to derive a composite term ($\gamma \times f$) as described by Mielke et al. (2013).

The analysis assumes that, during the period over which data are analysed, the relevant properties of the air mass are conserved and the losses of either measured species are not significant. It also assumes that the efficiency of $N_2O_5$ uptake and $ClNO_2$ / $NO_3^-$ production is independent of particle size. Later we examine the effect of considering uptake to coarse and fine aerosol particles separately as previously done by Mielke et al. (2013). Two further assumptions are 1) that measurement of $pNO_3^-$ accounts for the total production of nitrate by R3, i.e. that $NO_3^-$ formed from uptake of $N_2O_5$ does not significantly degas from the particle as $HNO_3$ and 2) that the formation of particulate nitrate via the net uptake of $HNO_3$ to aerosol as the temperature drops during the night is insignificant compared to that formed in $N_2O_5$ uptake. Note that the degassing of $NO_3^-$ as $HNO_3$ will result in an underestimation of $\gamma$, and overestimation of $f$, whereas the net uptake of $HNO_3$ (forming particulate nitrate) will have the opposite effect.

Regarding assumption 1) we note that the increase of particle nitrate during night is accompanied by an equivalent increase in ammonium (see Fig. 4) as gas-phase ammonia repartitions to form ammonium nitrate and buffers the release of $HNO_3$. For a given ammonia concentration, as the temperatures during nighttime fall, the partitioning increasingly favours the





retention of particulate nitrate over release of $HNO_3$. During a subsequent campaign at this site, we showed that the nighttime formation of particulate nitrate is dominated by $N_2O_5$ uptake (Phillips et al., 2013a). We also showed that, once corrected for the contribution of $N_2O_5$, the measurements of gas-phase $HNO_3$ did not reveal nighttime increases coincident with those of particle nitrate, suggesting that $N_2O_5$ uptake is not an important source of $HNO_3$ at this site. This is likely to be related to the cool nights at altitude of > 800 m and the abundance of ammonia in this mixed rural / industrialised region of Germany. Additionally, the strong temperature dependence of partitioning between ammonia, $HNO_3$ and ammonium nitrate would result in a bias to high values of $\gamma$ at low temperatures which prevent release of nitrate from the particles. The uptake coefficients obtained in this study showed no significant dependence on temperature (see later).

Regarding assumption 2) we note that, the strong correlation observed between $ClNO_2$ and $NO_3^-$ concentrations is a useful indicator that the source of particulate nitrate is dominated by $N_2O_5$ uptake and not $HNO_3$ uptake as $N_2O_5$ and $HNO_3$ was completely different diel profiles and atmospheric lifetimes.

Three different types of analysis were used to derive uptake coefficients and the fractional formation of $ClNO_2$. In the simplest method (1a) to derive $f$ we use longer time periods (several hours or the whole night) where plots of $[ClNO_2]$ versus $[NO_3^-]$ are approximately linear and values of $f$ are estimated from a linear fit of the data as exemplified in Fig. 5. On some nights (four in total) a linear relation is observed (Fig. 5a) whereas air mass changes can result in two distinct slopes (Fig. 5b, two nights in total) or such variability that analysis over a prolonged period is impossible (Fig. 5c, all other nights). For the example displayed in Fig. 5a a slope of $56.3 \pm 2.2$ pptv $\mu g^{-1}$ $m^3$ (2 $\sigma$ statistical error) results in a value of $f = 0.24 \pm 0.07$, where the error in $f$ includes overall uncertainty in the measurement of $ClNO_2$ and $NO_3$. Using this method, a total of 8 values of $f$ were obtained during the campaign.

In order to derive $\gamma$, absolute production rates of $NO_3^-$ and $ClNO_2$ are required and in method 1b shorter periods of data, usually between 1 and 3 hours, were chosen by inspection of the time series such that $NO_3^-$ and $ClNO_2$ concentrations both increase during a period of relatively constant composition and environmental variables such as temperature and RH. It is more likely that the assumptions hold during the shorter time periods chosen for 1b than the longer sections of data used in 1a. In this case, values of $pClNO_2$ and $pNO_3^-$ and average values of $A$ and $[N_2O_5]$ are calculated for the same period and inserted into Eqs. 5 and 6 to derive $f$ and $\gamma$. An example of this analysis is shown in Fig. 6, which indicates (grey shaded area) time periods in which $ClNO_2$ and $NO_3^-$ concentrations increased whilst other parameters (e.g. $N_2O_5$, $A$ and RH) were relatively constant. In this example between ~21:00 and 23:20 on the night 29.08-30.08, values of $pClNO_2$ ($14.6 \pm 3.0$ pptv/hr), $pNO_3^-$ ($21.0 \pm 5.1$ pptv /hr) , $\bar{A}$ ($7.6 \pm 2.2$) $\times 10^{-7}$ $cm^2$ $cm^{-3}$, $\overline{[N_2O_5]}$ ($153 \pm 43$ pptv) were obtained. The uncertainties quoted were derived by propagating statistical uncertainty (e.g. in the production rate of $ClNO_2$ from fitting to the data) and absolute uncertainty in the measurements of the concentrations of $ClNO_2$, $N_2O_5$, particulate nitrate and aerosol surface area as listed in section 2.2. Generally, the absolute error in the concentration measurements dominates the overall uncertainty for each parameter, the major exception being $N_2O_5$ which could be sufficiently variable over the averaging period for this to also contribute significantly. Inserting this set of parameters into Eqs. 5 and 6 results in $f = (0.82 \pm 0.26)$ and $\gamma = (7.3 \pm 3.1) \times$



$10^{-3}$ at RH = 77 ± 2 % and a temperature of 8 ± 1°C. Using this method a total of 12 values of γ were obtained during the campaign.

A more rigorous analysis, which avoids use of average values of $A$ and [$N_2O_5$] was also carried out. In this case $pNO_3^-$ and $p$ClNO$_2$ are calculated from 10 min averaged datasets using the right-hand sides of Eq. 3 and 4 respectively, taking measured values of $N_2O_5$ and $A$ at each time step and using an initial estimate for $f$ and γ. The predicted concentrations of ClNO$_2$ and NO$_3$ were then calculated for each time step by integration over the analysis time period. Values of $f$ and γ were then varied and the integration repeated until good agreement between observed and calculated [ClNO$_2$] and [NO$_3^-$] was obtained. An example of this type of analysis (for the night 19.08-20.08) is displayed in Fig. 7. The red lines are the results of the analysis in which the total particle surface area was used. Later, we discuss the effects of separately calculating the formation of NO$_3^-$ and ClNO$_2$ from the fine and coarse aerosol fractions. Unlike method 1b, a limitation of this procedure is that the analysis is limited to the first period of the night. Similar to method 1b it cannot predict negative changes in concentrations of ClNO$_2$ or NO$_3^-$ which are a result of changes in air-mass origin / age.

The values of $f$ and γ derived from method 1a-1c are plotted versus RH in Fig. 8. The data are colour coded according to the method used.

The lower panel of Fig. 8 reveals high variability in the values of $f$ obtained which range from 0.035 ± 0.027 to 1.38 ± 0.60. The errors on each individual determination vary from ~35 to ~100 %, which is the result of scatter in the data as well as uncertainty associated with measurement of ClNO$_2$ and particle-NO$_3^-$ and can result in non-physical values larger than unity. A further possible reason for values of $f$ that exceed unity is the degassing of NO$_3^-$ as HNO$_3$ from the particles, though as described above, this effect is expected to be small.

The average value of γ obtained for the campaign considering both methods 1b and 1c was 0.027 with a standard deviation of 0.03. Minimum and maximum values were 0.004 and 0.11, respectively. The large standard deviation reflects the great variability in the values of γ obtained, with up to a factor of 10 difference in γ at the same relative humidity. Considering the large variability, the values of γ obtained using methods 1b and 1c over the same time period are in reasonable agreement with an average ratio of 1.7. We compare the values of γ obtained during PARADE with other ambient datasets below.

## 4.2 Method 2. NO$_3$ steady state lifetime analysis

Atmospheric N$_2$O$_5$ is formed in a series of oxidation steps initiated mainly by the reaction of NO$_2$ with O$_3$ (R7).

$$NO_2 + O_3 \rightarrow NO_3 + O_2 \tag{R7}$$

$$NO_2 + NO_3 + M \rightarrow N_2O_5 + M \tag{R8}$$

$$N_2O_5 + M \rightarrow NO_2 + NO_3 + M \tag{R9}$$

Ambient concentrations of NO$_3$ and N$_2$O$_5$ are thus coupled via the gas-phase, thermochemical equilibrium that exists due to R8 and R9, so that the relative amounts of NO$_3$ and N$_2$O$_5$ are determined by temperature and NO$_2$ levels.





The so called "steady state" determination method for γ is based on the assumption that, after a certain period of time following sunset (often on the order of hours) the direct and indirect losses of $NO_3$ and $N_2O_5$ balance their production. The steady-state lifetimes can then be calculated from observations of the $NO_3$ and $N_2O_5$ concentrations and the production term $k_7[NO_2][O_3]$, where $k_7$ is the rate constant for reaction R7. This method was first used by Platt and co-workers to assess the heterogeneous loss of $N_2O_5$ (Platt and Heintz, 1994; Platt and Janssen, 1995; Heintz et al., 1996), and has been extended by Brown and co-workers to derive γ in regions distant from NO*x* sources such as the marine environment (Aldener et al., 2006) and aloft (Brown et al., 2006; Brown et al., 2009) and most recently for the continental boundary layer (Brown et al., 2016a). $[NO_2]$ and $[O_3]$ measurements are required to calculate the rate of $NO_3$ production, which is generally assumed to be via R7 only, though a contribution of $NO_2$ oxidation by stabilised Criegee intermediates has recently been hypothesised to represent a potential bias to this calculation (Sobanski et al., 2016). The steady-state analysis does not require any information about products formed by $N_2O_5$ heterogeneous reactions.

The inverse steady-state lifetimes of $NO_3$ ($\tau_{NO3}$) and $N_2O_5$ ($\tau_{N_2O_5}$) are given by expressions Eq.7 and Eq.8, respectively:

$$(\tau_{NO3})^{-1} \approx \gamma\left(0.25\bar{c}AK_{eq}[NO_2]\right) + k_g \tag{7}$$

$$(\tau_{N_2O_5})^{-1} \approx k_g\left(K_{eq}[NO_2]\right)^{-1} + 0.25\bar{c}A\gamma \tag{8}$$

Where $K_{eq}$ is the temperature dependent equilibrium constant describing the relative concentrations of $NO_2$, $NO_3$ and $N_2O_5$ (R8 and R9), $[NO_2]$ is the concentration of $NO_2$, and $k_g$ is the pseudo first-order loss constant for $NO_3$ loss in gas-phase reactions (e.g. with NO or with hydrocarbons). A plot of $(\tau_{NO3})^{-1}$ or $(\tau_{N_2O_5})^{-1}K_{eq}[NO_2]$ against $0.25\bar{c}AK_{eq}[NO_2]$ should be a straight line with γ as slope and $k_g$ as intercept. This method thus relies on the fact that the relative concentrations of $N_2O_5$ and $NO_3$ vary with $K_{eq}[NO_2]$ and thus the contribution of their individual losses to the overall lifetime of both $NO_3$ and $N_2O_5$ also varies with $[NO_2]$ once changes in temperature (and thus $K_{eq}$) are accounted for. The method therefore assumes that, for a given analysis period in which $NO_2$ is changing sufficiently to change the relative loss rates of $NO_3$ and $N_2O_5$, both γ and $k_g$ are constant (i.e. do not depend on $[NO_2]$). This will not always be the case and we have often observed that the relation between $(\tau_{NO3})^{-1}$ and $K_{eq}[NO_2]$ is non-linear. In environments where $NO_3$ losses are dominated by gas-phase reactions of $NO_3$, the uncertainty associated with the derivation of γ via a steady-state analysis is clearly greatly enhanced.

Figure 9 illustrates some of these issues for data obtained over a period of several hours on the nights 1-2.09, 2-3.09 and 5-6.09. The first two nights (1-2.09 and 2-3.09) had long $NO_3$ lifetimes and were selected for analysis as long $NO_3$ lifetimes represent periods with low rates of $NO_3$ loss by gas-phase processes (i.e. $k_g$ is small). We have previously shown (Sobanski et al., 2016) that the long $NO_3$-lifetimes on these nights result from sampling from a low altitude residual layer. A large variability in the nighttime $NO_2$ mixing ratio on these nights (~1 to 13 ppbv, see Fig. 9a and 9c) should result in a significant shift in the $NO_2$-to-$N_2O_5$ ratio and thus sensitivity in the $NO_3$ lifetime to the uptake of $N_2O_5$ to aerosol. On the night 1-2[nd] Sept., plume like features in the $NO_2$ mixing ratio at ~22:00, 01:30 and 04:00 were accompanied by decreases in the steady-state $NO_3$ lifetime. The inverse $NO_3$ lifetime is plotted against $0.25\bar{c}AK_{eq}[NO_2]$ in Fig. 9b. Here we have selected data that



was obtained from about 2 hr after sunset to the next dawn when $NO_3$ started to decrease as represented by the red data points in Fig. 9a. The slope of the plot results in values of $\gamma = (9.6 \pm 0.7) \times 10^{-3}$ and $k_g = (2.0 \pm 0.6) \times 10^{-4}$ s$^{-1}$, where the errors are statistical only. Over the period analysed, the relative humidity (blue line in Fig. 9a) was $60 \pm 2$ %.

On the night 2-3.09 at about 19:30 UTC ($\approx$ 1 hour after sunset), the $NO_3$ lifetime increases gradually to a value of $\sim$ 900 s

until 21:30 as $NO_2$ decreases from $\sim$ 7 to 3 ppbv. A rapid increase in $NO_2$ at 21:30 is then accompanied by a much shorter $NO_3$ lifetime. After $\sim$22:00, $NO_2$ slowly decreases and the $NO_3$ lifetime recovers to about 500 s. Thus, also in this night there is a clear anti-correlation between $NO_2$ and the $NO_3$ lifetime. Figure 9d plots the inverse of the $NO_3$ lifetime versus $0.25\bar{c}AK_{eq}[NO_2]$ with the data point colour coded according to relative humidity. The first period of the night (orange, yellow and green data-points) are best described (black line) by an uptake coefficient of $(2.7 \pm 0.2) \times 10^{-2}$ and $k_g = (1.0 \pm$

$0.4) \times 10^{-4}$ s$^{-1}$. The second period (starting $\sim$ 5 hours after sunset) is best characterised by larger values of $\gamma = (4.5 \pm 0.3) \times 10^{-2}$ and $k_g = (1.5 \pm 0.1) \times 10^{-3}$ s$^{-1}$ (errors are statistical only).  We note that prior to the peak in $NO_2$ at 21:30 the relative humidity of the air was stable at $\approx 70 \pm 3$ % whereas after 22:00 it remained at $80 \pm 3$ %. A shift in air mass to one with larger water vapour content could help explain the larger values of $\gamma$ obtained in the 2$^{nd}$ period of this night and also may also be the reason for the larger gas-phase losses of $NO_3$ if the more humidified air mass is more impacted by boundary-layer

emissions. On two other nights when the $NO_3$ lifetimes were long (30-31.08 and 31.08-1.09) there was very little variation in $NO_2$ so that the steady-state lifetime of $NO_3$ was insensitive to the uptake coefficient.

In Fig. 9e and 9f we present data that were obtained with sufficient variation in $NO_2$, but with relatively short $NO_3$ lifetimes. As in the datasets discussed above, $NO_3$ lifetimes are anti-correlated with $NO_2$. However, the plot of the inverse of the $NO_3$ lifetime versus $0.25\bar{c}AK_{eq}[NO_2]$ in Fig. 9f has a very large slope, resulting in $\gamma \approx 0.3$ and also a negative intercept.

Unrealistically large values of $\gamma$ and negative intercepts can be caused by a covariance between $NO_2$ concentrations and gas-phase losses of $NO_3$, which makes this type of analysis unviable. Similar observations of apparently negative gas-phase reactivity indicating breakdown of the steady-state method have previously been made (Morgan et al., 2015; Brown et al., 2016a).  In addition to co-variance between $NO_2$ and other trace-gases that are reactive to $NO_3$, negative intercepts can also result from analysis of time periods in which steady state was not acquired. During the PARADE campaign there were only

three clear examples of the expected relationship between the $NO_3$ lifetime and $NO_2$ mixing ratios as defined by Eq. 7 or 8. This indicates that, over periods of a few hours, the changes in the $NO_3$ lifetime observed cannot be attributed solely to different rates of heterogeneous processing but that other air mass characteristics (e.g. influencing the gas-phase losses of $NO_3$) are also variable, which may be expected in a region that is impacted by fresh emissions of reactive gases. We also note that the average aerosol surface area during the PARADE campaign was about 70 $\mu$m$^2$ cm$^{-3}$, which is low compared to

the ones reported e.g. by Aldener et al. (2006), Brown et al. (2006) and Brown et al. (2009) which range from 200 to 600 $\mu$m$^2$ cm$^{-3}$ and where plots of inverse $NO_3$ lifetime versus $0.25\bar{c}AK_{eq}[NO_2]$ resulted in straight lines and returned reasonable values of $\gamma$. We conclude that the steady-state approach to derive $\gamma$ is best suited to air masses with high aerosol loading but remote from fresh emissions and only worked on a few occasions during the PARADE campaign when $NO_3$ was sampled





from the residual layer. The overall uncertainty associated with the determination of γ via the steady-state method is derived from uncertainty in the mixing ratios of $O_3$ (5 %) and $NO_3$ (25 %), the aerosol surface area (~70 %), the rate constant for reaction between $O_3$ and $NO_2$ (15 % at 298 K (Atkinson et al., 2004)) and the equilibrium constant, $k_{eq}$ (20 % at 298 K, (Burkholder et al., 2016)) and when propagated in quadrature is equal to ± 75 %. The three values of γ obtained by the

steady-state method (with total uncertainty) are also plotted in Fig. 8.

### 4.3 Method 3. Iterative box model of $N_2O_5$ formation and loss.

A further method to derive $N_2O_5$ uptake coefficients is to use a box model of $NO_3$ / $N_2O_5$ formation and loss (both gas-phase and heterogeneous) with γ treated as a variable to match observed $N_2O_5$ concentrations. Such an analysis was conducted by Wagner et al. (2013) who used an iterative box model of $N_2O_5$ chemistry to estimate the uptake coefficient that would be

needed to account from the $N_2O_5$ concentration measured at the observation point at time "t" after sunset. The calculations were constrained by on-site measurements of $N_2O_5$ precursor gases ($NO_2$ and $O_3$) which were used to estimate the original $NO_2$ and $O_3$ concentrations (i.e. at sunset) for the same air mass, and measurements of the aerosol surface area. As the concentration of $N_2O_5$ at any time after sunset depends not only on its heterogeneous loss processes but also on gas-phase losses of $NO_3$, the total $NO_3$ reactivity must also be known. As the authors point out, this can be a poorly constrained

parameter as potentially not all reacting VOCs are measured and the reactions of $NO_3$ with other trace gases including NO and $RO_2$ during transport to the measurement site cannot be accurately assessed. This method is therefore expected to be least accurate when gas-phase losses of $NO_3$ are a substantial fraction of the overall loss rate of $NO_3$ and $N_2O_5$. During much of the PARADE campaign (and also for previous measurements at this site (Crowley et al., 2010) $NO_3$ lifetimes were dominated by gas-phase losses and on the few occasions when $NO_3$ was long-lived this was due to sampling from a residual

layer that was decoupled from boundary layer emissions. As we have recently shown (Sobanski et al., 2016) when sampling from the residual layer, measured VOC mixing ratios at the site are incompatible with the high $NO_3$ and $N_2O_5$ levels observed and a box-model approach that requires a constraint on gas-phase losses of $NO_3$ cannot provide reliable results. For this reason, we have not attempted to analyse our data in a similar fashion to Wagner et al. (Wagner et al., 2013).

### 4.4 Comparison of γ and $f$ with literature values derived from ambient datasets

Previous determinations of γ from ambient datasets are summarised along with the results of this work in Table 1. The reported values of γ vary over more than two-orders of magnitude, between 0.0005 to ~0.1. The directly measured loss rates of $N_2O_5$ to ambient aerosol (Bertram et al., 2009b; Riedel et al., 2012b) convert to variable values of γ (~0.005 – 0.035) in Seattle (RH ~ 74 ± 13 %) that are comparable to those reported in this work. The largest values of γ were obtained at low organic to sulphate ratios, whereas their largest values at a coastal location impacted by chloride containing aerosol were





obtained when the molar $H_2O$ / $NO_3^-$ ratio was high. In drier conditions in Boulder (RH < 30 %) γ was much lower (< 0.005) and independent of the organic to sulphate ratio.

Apart from the present study, the steady state method using $NO_3$ lifetimes has been successfully used for analysis of aircraft data (Brown et al., 2006; Brown et al., 2009; Morgan et al., 2015), ship data (Aldener et al., 2006) and also in ground-based

studies in which residual layer air was sampled (Brown et al., 2016a). In their airborne studies over the NE U.S., Brown et al. (2006) report regional differences in γ which they assign to changes in the sulphate content or sulphate-to-organic ratio of the particles. In a further airborne study over Texas, Brown et al. (2009) analysed several residual layer plumes to derive ~ 30 values of γ between $5 \times 10^{-4}$ and 0.006 during three flights. They found no dependence of γ on RH or aerosol composition, which was typically ammonium sulphate with an organic fraction of > 50 %. In contrast, airborne

measurements over Europe (Morgan et al., 2015) suggest that γ is larger and dependent on the particulate nitrate content. Measurements of $NO_3$ and $N_2O_5$ on a polluted mountain site in Hong Kong (Brown et al., 2016b), enabled 10 derivations of γ which varied between 0.004 and 0.029. Measurements of $ClNO_2$ at the same location indicate that $N_2O_5$ uptake was, at least partially, to chloride containing particles. Values of (0.03 ± 0.02) for γ obtained over the ocean (Aldener et al., 2006) are, on average, at the higher range of ambient measurements, potentially reflecting the role of particulate chloride.

In their analysis of $N_2O_5$ and $ClNO_2$ datasets obtained during CalNex-LA at a coastal location, Mielke et al. (2013) did not derive separate values of $f$ and γ but report a composite term, $\gamma f$. For submicron particles their values of $\gamma f$ vary ~ two orders of magnitude during a single night with campaign minimum values close to $10^{-4}$ and maximum values of 0.05. In comparison, $\gamma f$ values from the present study vary between ~0.001 and 0.09. Mielke et al. (2013) report a campaign average value of $\gamma f$ = 0.0084, which is similar to the average value of $\gamma f$ = 0.014 obtained in the present study. Similar values may

indeed be expected despite the different locations (semi-rural mountain site in PARADE versus polluted coastal in CalNex-LA) as most of the γ values reported here were derived using $ClNO_2$ observations, i.e. for particles with chloride content as would certainly be expected at the coastal location.

Wagner et al. (2013) used a box-model and observations of $N_2O_5$ to derive highly scattered values of γ with two peaks in the frequency distribution at 0.015 and 0.04, lower values of γ being associated with higher particulate nitrate content. As

discussed above, this method to derive γ requires knowledge of the $NO_3$ lifetime with respect to gas-phase losses, and the authors suggest that uncertainty in this parameter may result in values of γ that are too high. The source of the very high scatter in γ derived by this method is unlikely to be related to aerosol composition, but probably results from the large variability of $N_2O_5$ (and $NO_3$) frequently observed in ground based measurements, which is the result of sampling advected air masses with strong vertical gradients in $NO_3$ in a poorly mixed boundary layer at night.

In summary, the derivation of γ from ambient datasets reveals great variability in the values obtained, with occasional evidence for a suppressing role of particulate organics and nitrate, which is consistent with laboratory observations. Below we examine our dataset for evidence of such effects and compare the γ values obtained to parameterisations.



The large spread in $f$ derived in the present study is consistent with previous analyses of field measurements in which values between < 0.01 and > 0.9 have been reported (Riedel et al., 2012b; Wagner et al., 2013; Young et al., 2013). The large spread in $f$ is to be expected as this parameter is controlled largely by particulate chloride content, though, as mentioned already, some dissolved organic species my compete with $Cl^-$ for reaction with $NO_2^+$ and thus reduce the $ClNO_2$ yield for a given chloride content. We thus expect $f$ to be largest in polluted coastal regions and lowest (or zero) in continental regions with no marine influence or anthropogenic chlorine emissions.

### 4.5 Factors affecting the $ClNO_2$ formation efficiency, $f$

From the PARADE dataset, we derived values of $f$ that ranged from $0.029 \pm 0.027$ to $1.38 \pm 0.60$, with a mean value of 0.49. To put these numbers into context, we rearrange Eq.2 and use a $k_2 / k_4$ ratio of $2.2 \times 10^{-3}$ to calculate which particulate chloride concentrations would be required to explain the values of $f$ observed (Eq. 7).

$$[Cl^-] = 2.2 \times 10^{-3} [H_2O] / (f^{-1} - 1) \tag{9}$$

Assuming that the fate of $H_2NO_3^+$ is reaction with $Cl^-$ and $H_2O$ and taking a water concentration of ~40 M, we obtain $f$ ~ 0.035 (at the lower limit of values obtained) when $N_2O_5$ is taken up to particles with $[Cl^-] = 3.4 \times 10^{-3}$ M. For $[Cl^-] = 0.85$ M, $f$ ~ 0.9. Chloride concentrations exceeding 1 M may be found in sea-salt particles, which have [Cl] close to 5 M when freshly generated (Sander and Crutzen, 1996). While the sodium content of the particles is conserved during transport, there are mechanisms by which chloride may be lost to the gas-phase including acid-displacement of HCl following uptake of $HNO_3$ or $H_2SO_4$ and also reactive losses via uptake of $N_2O_5$ and other inorganic trace gases and radicals (Keene et al., 1990; Keene et al., 1999; von Glasow et al., 2001). The presence of chloride in non-marine aerosol is related to the uptake of HCl to particles and formation of ammonium chloride, where the HCl may be either of anthropogenic or marine origin. As discussed by Phillips et al. (2012), there is indirect evidence for the presence of aged, sea-salt aerosol at the Kleiner Feldberg site during the PARADE campaign, which is largely based on the formation of $ClNO_2$ on air mass origin as derived from back trajectories and wind-direction measurements. During a campaign (INUIT, (Phillips et al., 2013a)) at the same site and time of year in 2012, we measured the inorganic particle composition via ion-chromatography and showed that the site is regularly impacted by marine aerosol during periods of strong north-westerly winds. During INUIT, high concentrations of particulate inorganic chloride (up to ~2 µg m$^{-3}$) were strongly correlated with sodium (up to ~1.6 µg m$^{-3}$) with a chloride / sodium molar mixing ratio that was significantly lower than unity, indicating loss of chloride from sea-salt during transport from the coastal source regions (~ 400 km distant). For the purpose of illustration, 1000 deliquesced particles cm$^{-3}$ with an average diameter of 300 nm and a 0.85 M chloride concentration (as derived from our measurement of $f$) would result in ~ 0.4 µg m$^{-3}$ [Cl$^-$] suggesting that the high values of $f$ obtained are compatible with chloride particle concentrations at this site, albeit not measured simultaneously. A high efficiency of $ClNO_2$ formation $f$ (> 0.5) was measured in the period between 29.08 and 03.09 for which two-day back trajectories indicate that the site was influenced by marine air near the UK (Phillips et al., 2012). Over this period, NR PM1 Cl$^-$ increases on a number of occasions in concert with nocturnal $ClNO_2$ production.





The lower values of $f$ derived (e.g. 0.035) are associated with mM [Cl⁻] particle concentrations and are likely the result of $N_2O_5$ uptake to non-marine particles to which HCl has partitioned during transport.

### 4.6 Factors affecting the $N_2O_5$ uptake efficiency, $\gamma$

The average value of γ derived from the PARADE dataset from methods 1b, 1c and steady-state analysis is 0.028 with a
large standard deviation (0.029) reflecting the high variability in this parameter. No significant dependence on temperature was observed (Fig. 10).

The values of γ derived from the PARADE dataset are compared in Fig. 8 to those predicted from different parameterisations available in the literature, all derived from laboratory studies. The parameterisation listed by the IUPAC panel considers, via the resistor model, the dependence of γ on the bulk accommodation coefficient ($\alpha_b$), the solubility ($H$) and diffusivity ($D_l$) of
$N_2O_5$ and the rate coefficient, $k_{H2O}$, for its hydrolysis (Eq. 8).

$$\gamma_{\text{IUPAC}} = \left\{ \frac{1}{\alpha_b} + \frac{\bar{c}}{4HRT(D_l k_{\text{H}_2\text{O}}[\text{H}_2\text{O}])^{0.5}} \right\}^{-1} \qquad (10)$$

IUPAC preferred values are listed (for ammonium sulphate) as $\alpha_b = 0.03$, $k_{H2O} = 1.0 \times 10^5$ M⁻¹s⁻¹, $D_l = 1 \times 10^{-5}$ cm² s⁻¹ and H = 2 M atm⁻¹.

The more complex parameterisation of Bertram and Thornton (2009) considers the concentrations of particulate nitrate, [$NO_3^-$], chloride, [Cl⁻], and water [$H_2O$] (Eq. 9).

$$\gamma_{\text{BT}} = Ak_1 \left( 1 - \frac{1}{\left( \frac{k_2[\text{H}_2\text{O}(l)]}{k_{-1}[NO_3^-]} \right) + 1 + \left( \frac{k_4[Cl^-]}{k_{-1}[NO_3^-]} \right)} \right) \qquad (11)$$

where $A = 3.2 \times 10^{-8}$ s, $k_1 = 1.15 \times 10^6 - 1.15 \times 10^6 \exp(-0.13[\text{H}_2\text{O}(l)])$ s⁻¹, $k_2 / k_{-1} = 0.06$ and $k_4 / k_{-1} = 29$ as empirically derived
and listed in Bertram and Thornton (2009). We refer to the calculated values of γ as $\gamma_{IUPAC}$ and $\gamma_{BT}$, respectively.

Other parameterisations have been developed which also deal with the effects of particle organics e.g. as a particle coating (Anttila et al., 2006) and have been reviewed by Chang et al (Chang et al., 2011). We examine the effects of assuming that a hydrophobic coating covers the particles below.

For the parameterisation of Bertram and Thornton (2009) (henceforth referred to as BT), particulate nitrate, sulphate and
ammonium concentrations as measured by the AMS and the relative humidity were then used as input parameters to calculate the particle liquid water content [$H_2O(l)$] using the E-AIM model (http://www.aim.env.uea.ac.uk/aim /model3/model3a.php) (Clegg et al., 1998; Wexler and Clegg, 2002). The particulate chloride content was calculated using values of $f$ derived as described above and in Eq. 2. In those cases where $f$ exceeded unity, it was set to 1.0 for calculation of particulate chloride content (Eq. 7) to avoid generation of negative concentrations. For the γ values obtained using the
steady-state method, the chloride content is unknown and was set to zero. Given insufficient information on the identity of





the condensed organics or particle hygroscopicity, the influence of particle organic content on the particle water was not considered.

$\gamma_{BT}$ (red stars) and $\gamma_{IUPAC}$ (blue stars) were computed for each observational data point in Fig. 8. On average, the predictions and the measurements are in reasonable agreement. The variability in the $\gamma_{BT}$ predicted values (for a given RH) stems from

different particulate chloride content. As an example of the sensitivity to chloride, the low value predicted for 70 % RH in Fig. 8 was obtained for a calculation of uptake to particles which had a high nitrate content and, as the chloride content was not known for this particular period ($\gamma$ was derived using the steady-state method), it was set to zero. Adding ~0.02 M chloride to the calculation (equivalent to $f \sim 0.2$) would increase the calculated $\gamma$ by ~40 % (from 0.007 to 0.01). The BT-predicted $\gamma$, averaged over the same time periods as the measured values, is ($0.028 \pm 0.008$). This is entirely consistent with

the campaign averaged value derived from methods 1b and 1c of ($0.025 \pm 0.027$), as described above. However, the BT-parameterisation lies at the upper range of the measurements for relative humidities between 65 and 75 % and at the lower end for the highest relative humidities encountered. As the particles during PARADE have significant organic content (see Fig. 2) this may reflect the dependence of the organic suppression of $\gamma$ on relative humidity as reported by (Gaston et al., 2014).

As already stated, the BT-parameterisation accounts for a reduction in $\gamma$ due to particulate nitrate but does not take the effect of particle organic content into account. $\gamma$ values derived from observation of $N_2O_5$ uptake to ambient particles in Seattle (Bertram et al., 2009b) have revealed that the uptake coefficient can be dependent on the sulphur-to-organic ratio, with reduced uptake at low ratios. In Fig 10. we plot the measured uptake coefficients against the molar $H_2O$ / nitrate ratio of the particles and the organic-to-sulphate ratio. Despite significant variation in both nitrate content (~0.2 to 7 μg m$^{-3}$) and the

organic-to-sulphate ratio (0.3 - 1.2), no trend is seen. The PARADE $\gamma$s are nonetheless consistent with the Seattle observations reported by Bertram et al. (2009b) in which $\gamma$ was close to $3 \times 10^{-2}$ for organic/sulphate ratios up to 3. The large organic/sulphate ratios up to ~13 observed by Bertram et al. were not encountered during the PARADE campaign. As discussed by (Gaston et al., 2014), the effect of organic content in a particle will depend on the amount and oxidation state of the organics, with aged particles containing a high fraction of oxidised organics least impacting on $\gamma$, especially at high

relative humidities.

The lack of a measureable dependence on particulate nitrate content may be contrasted with the interpretation of Morgan et al. (2015), who report that the uptake coefficients they derived in their steady-state analysis of airborne data ($0.03 > \gamma > 0.007$) reveal a dependence on the $H_2O$ / nitrate ratio, as would be expected from Eq. 8, with the lower values of $\gamma$ associated with high nitrate content. The larger scatter in uptake coefficients in the present study would most likely disguise this effect.

In addition, the formation of $ClNO_2$ is an unambiguous confirmation of the presence of chloride in the particles, which will efficiently counter the reduction in $\gamma$ caused by nitrate. We also note that the effect of particulate nitrate content is difficult to assess for the PARADE campaign. As we have already shown for this site (Phillips et al., 2013a) a major fraction of the





particulate nitrate is in any case the result of $N_2O_5$ uptake, so that efficient uptake (i.e. high values of γ) may be accompanied by large nitrate concentrations, essentially implying a time dependence to γ as the uptake proceeds (and particulate nitrate increases) or, in other words, an apparent dependence of γ on $N_2O_5$ levels. While such effects are reported in laboratory experiments they will be very difficult to observe in ambient datasets and we see no dependence of γ on the $N_2O_5$ levels.

We have also examined the possibility that the formation of $ClNO_2$ occurs only in coarse particles, i.e. with > 0.5 μm diameter as measured by the APS. In this case, assuming the coarse particles to be sea-salt we fixed γ at 0.02. (Ammann et al., 2013). Similar to the conclusions of Mielke et al. (2013), we find that even when the $ClNO_2$ efficiency is optimised by setting and $f$ to 1, only a fraction (~20 %) of the $ClNO_2$ observed can be accounted for. This indicates that chloride is distributed across the particle size spectrum and, after 2-3 days of transport across polluted NW-Europe, is also in the form

of gas-phase HCl in equilibrium with ammonium chloride as well as in coarse particles as NaCl.

The presence of an organic coating can reduce the uptake coefficient of $N_2O_5$ to an aqueous particle and a parameterisation of γ based on laboratory studies has been developed to account for this (Mentel et al., 1999; Anttila et al., 2006; Riemer et al., 2009). In this case, the uptake coefficient is given by:

$$\gamma_{coating} = \frac{4RTH_{org}D_{org}R_{aq}}{\bar{c}LR_{part}} \tag{12}$$

Where $H_{org}$ and $D_{org}$ are the solubility and diffusivity, respectively of $N_2O_5$ in the organic coating (of thickness $L$). $R_{aq}$ and $R_{part}$ are the radii of the aqueous core and the particle, respectively. In order to calculate the thickness of the coating, we assume that the entire organic content of the particle is hydrophobic, has a relative density of 1.27 and forms a coating on an aqueous, inorganic core (of density 1.28 as returned from the E-AIM calculation). We set $H_{org}D_{org}$ to $0.03 \times H_{aq}D_{aq}$ as derived by Anttila et al. (2006) in their laboratory studies with $H_{aq}$ = 5 mol L$^{-1}$ atm$^{-1}$ and $D_{aq}$ = $10^{-9}$ m$^2$ s$^{-1}$. The net uptake to a particle

with an aqueous core (containing nitrate and chloride) and an organic coating is then given by:

$$\frac{1}{\gamma_{net}} = \frac{1}{\gamma_{BT}} + \frac{1}{\gamma_{coating}} \tag{13}$$

As shown in Fig. 8, the effect of the organic coating can be substantial and values of $\gamma_{net}$ obtained are significantly lower than those calculated from the BT-parameterisation and also the observations. This is not totally surprising, as a significant fraction of the organic content of an aged particle is likely to be soluble in water, which means that the assumed thickness of

the organic coating is an upper limit and the effect of organic in suppressing γ is overestimated (Gaston et al., 2014). The scatter in our data and the missing information regarding the chemical nature of the organics in the particle (to get realistic values of $H_{org}$ and $D_{org}$) do not allow us to further evaluate the role of organics in suppressing γ.

## 5 Conclusions

We present estimates of γ using ambient measurements of gas and particle composition at the Kleiner Feldberg observatory,

near Frankfurt, SW Germany, during the PARADE observational experiment in the summer of 2011. Values of γ were





derived using different methods and reveal high variability with ($0.004 < \gamma < 0.11$) and an average value of $0.028 \pm 0.027$. The results are compared with different parameterisations based on laboratory data and are in reasonable agreement when we neglect the potential effect of an organic coating on the particle but account for inorganic composition and relative humidity (Bertram and Thornton, 2009). The assumption that the organic fraction of the particle is in the form of a hydrophobic

coating (Anttila et al., 2006) results in predicted $\gamma$ values that are inconsistent with our dataset, and is clearly inappropriate for the aerosol encountered during the PARADE campaign.

*Acknowledgements*

We would like to thank the staff and department of the Johann Wolfgang Goethe-University, Frankfurt am Main for

logistical support and the use of the Taunus Observatory. We acknowledge the help and support of all PARADE participants and our colleagues in the Department of Atmospheric Chemistry, MPIC.

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



**Table 1: Values of γ derived from ambient datasets**

| Platform (location / height) | γ (method) | Notes | reference |
|---|---|---|---|
| Ship (East Coast U.S., 11 m) | 0.03 ± 0.02 (ss) | Polluted marine | (Aldener et al., 2006) |
| Aircraft (NE-U.S. < 1.5 km) | < 0.0016 - 0.017 (ss) | Continental / coastal residual layer. γ dependence on organic / sulphate content. | (Brown et al., 2006) |
| Aircraft (Texas, U.S. < 1 km) | 0.0005-0.006 (ss) | Continental / coastal residual layer. γ variable but independent of RH or aerosol composition | (Brown et al., 2009) |
| Ground (US, Seattle/Boulder 5-10 m) | < 0.01-0.04 (AFR) | Urban / sub-urban environment. γ dependence on organic / sulphate ratio and RH. | (Bertram et al., 2009b) |
| Ground (US, La Jolla, 15 m) | <0.001– 0.029 (AFR) | Polluted coastal environment. γ dependence on nitrate content | (Riedel et al., 2012b) |
| Ground (U.S, Boulder 10-300 m) | 0.002–0.1 (Box model) | Continental, pollution impacted boundary layer / residual layer. γ dependence on nitrate content | (Wagner et al., 2013) |
| Aircraft (N.W. Europe 500-1000 m) | 0.01–0.03 (ss) | Continental, pollution impacted residual layer / free troposphere. γ dependence on nitrate content | (Morgan et al., 2015) |
| Ground US, Pasadena (10 m) | $\gamma f$ = 0.008 (av) | Coastal (CalNex-LA). $\gamma f$ was suppressed by particle organic content and enhanced by particulate chloride content. | (Mielke et al., 2013) |
| Ground China, Hong Kong, (957 m) | 0.004–0.029 (ss) | Coastal, pollution impacted mountain site. | (Brown et al., 2016a) |
| Ground (S.W. Germany, 650 m) | 0.004–0.11 (1b, 1c, ss) | Semi-rural mountain site with anthropogenic influence, mixed boundary layer / residual layer | This work |

**Notes**: ss = steady-state analysis. AFR = aerosol flow reactor. Av = averaged over a campaign.



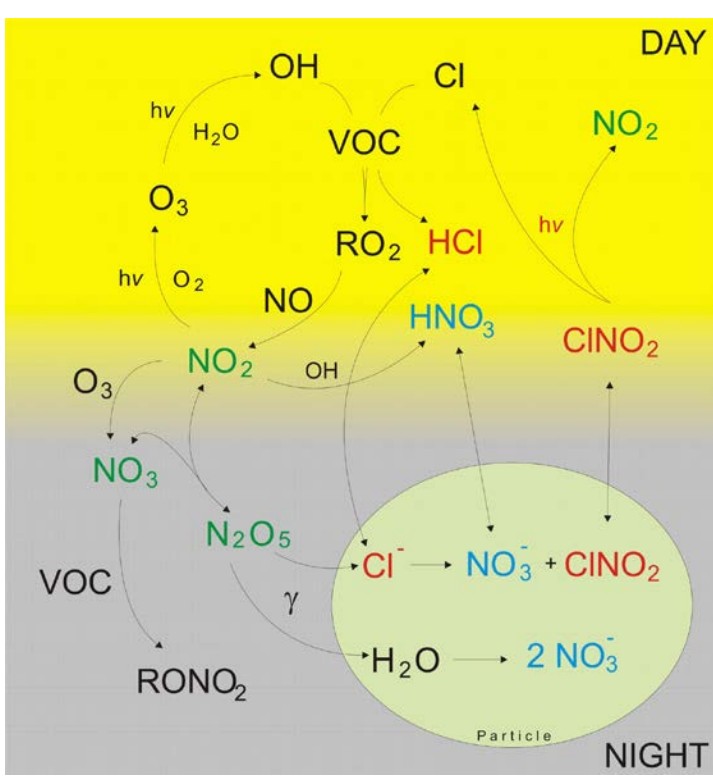

**Fig. 1**. Gas- and aqueous-phase chemical processes forming $ClNO_2$ and particulate nitrate indicating the role of $ClNO_2$ in modifying the effect of $N_2O_5$ uptake on the chemical lifetime of NO$x$ and photochemical $RO_2$ generation.



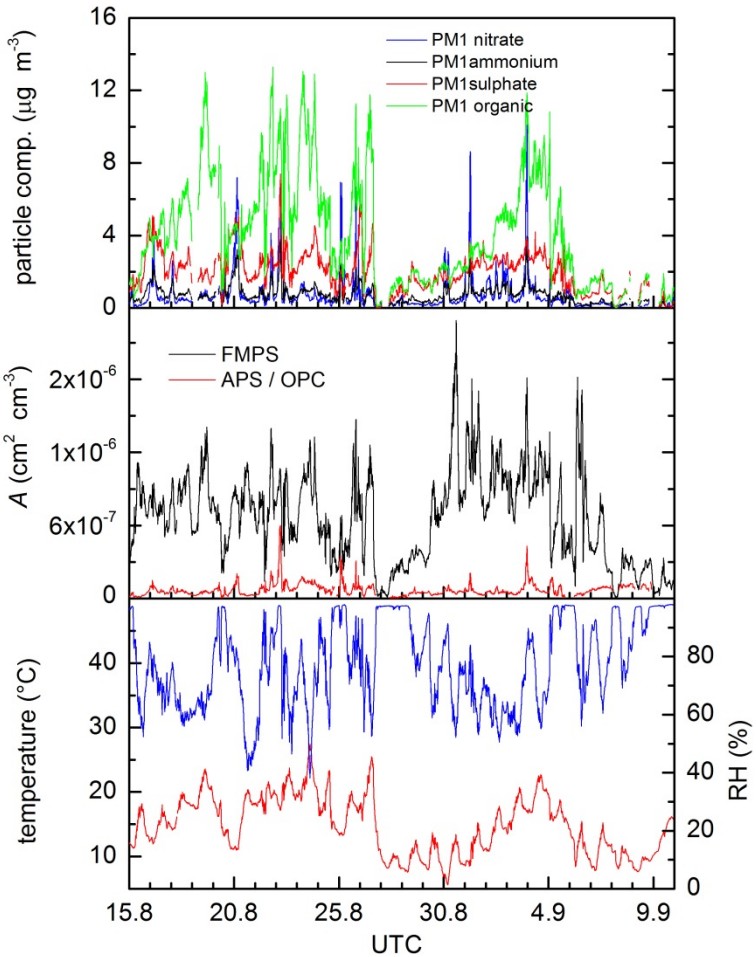

**Fig. 2.** Time series of temperature, relative humidity and particle properties during PARADE. *A* = aerosol surface area. Non-refractory, PM1 organic, nitrate, sulphate and ammonium were measured by the AMS. Aerosol surface area was measured by an FMPS (20-500 nm) and by APS (>0.5 μm) and OPC (>0.25 μm).



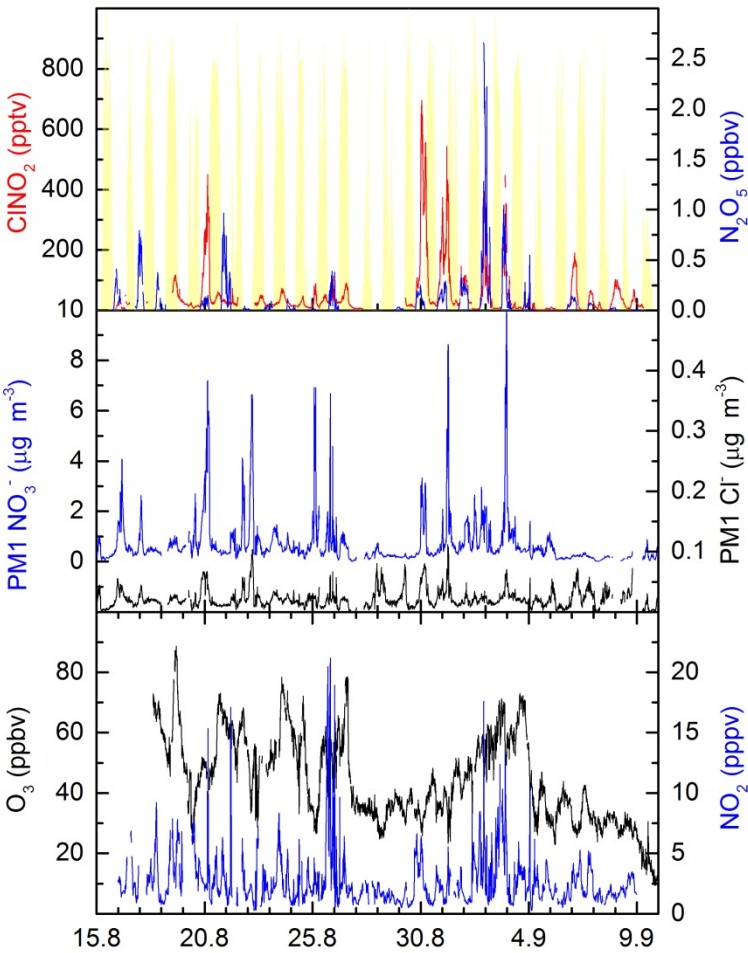

**Fig. 3**. Time series of ClNO$_2$ and its precursor, N$_2$O$_5$. Global radiation (yellow) is plotted to separate day-night periods and to indicate the day-to-day, relative photochemical activity. The precursors to N$_2$O$_5$ (NO$_2$ and O$_3$) are shown as are the AMS measurement of particulate nitrate (as in Fig. 2) and chloride.





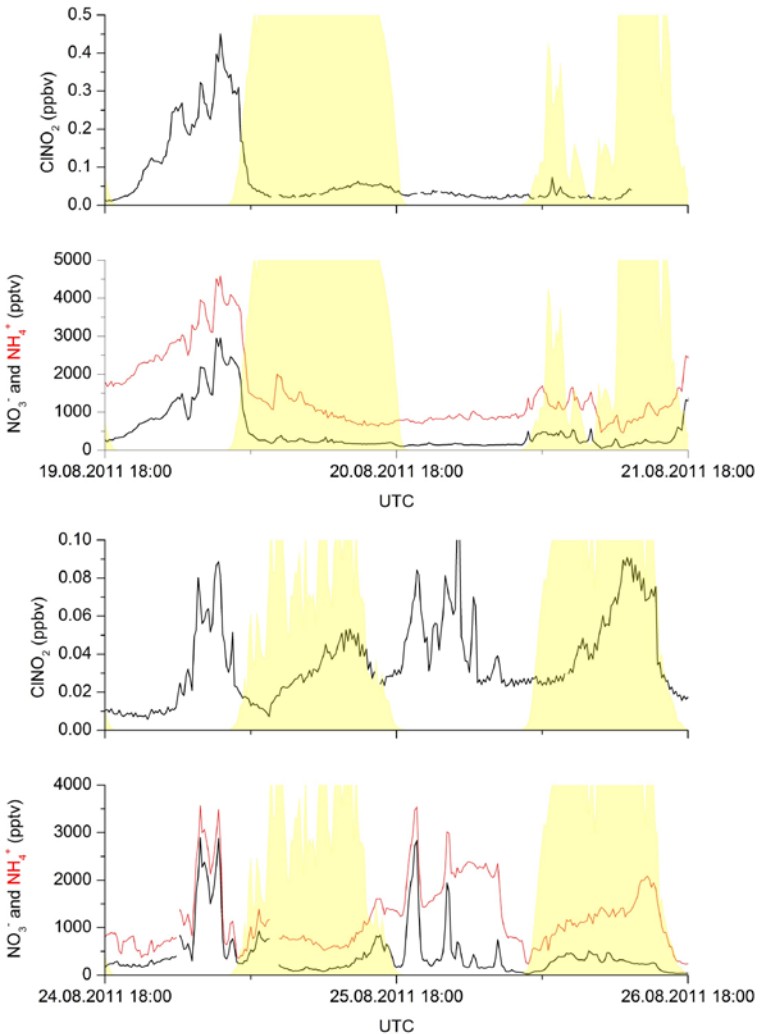

**Fig. 4**. Zoom-in on particle-nitrate and ClNO$_2$ formation on 4 nights. Global radiation (yellow) is plotted to separate day-night periods and indicate the day-to-day, relative photochemical activity. Note that the ClNO$_2$ plotted for these days was measured at $m/z = 161.9$ and therefore contains a contribution from HCl, hence the apparent daytime production.



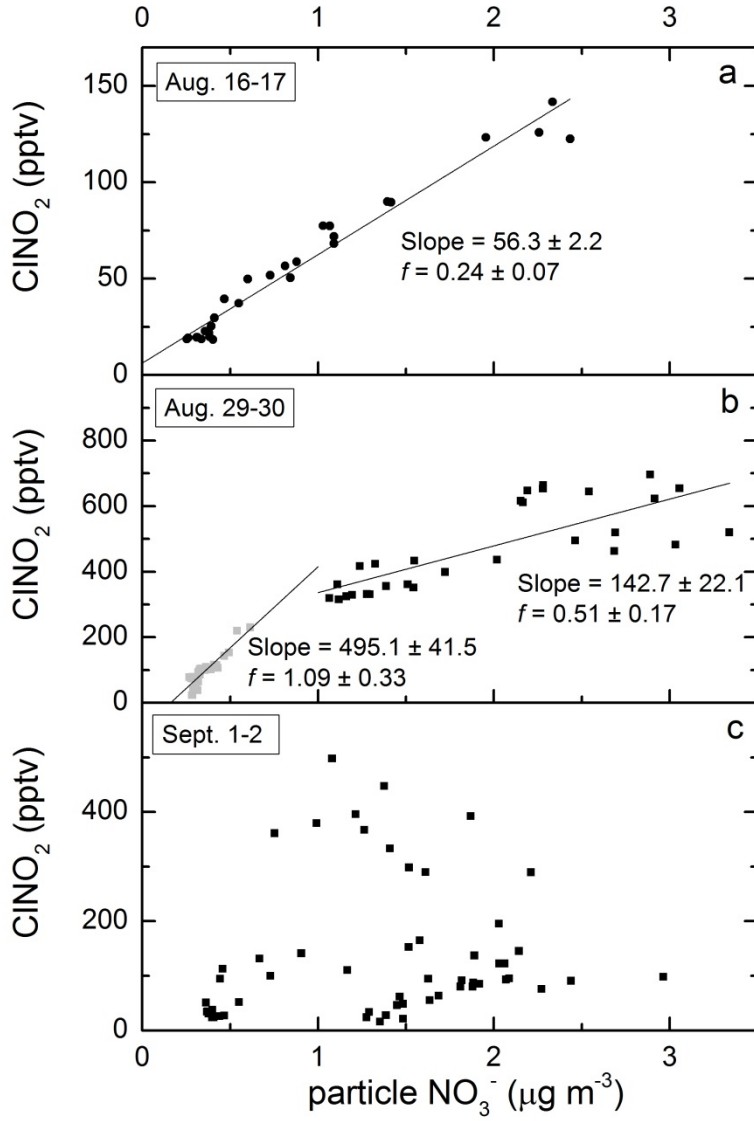

**Fig. 5**. Plots of $ClNO_2$ versus $NO_3^-$ for three different campaign nights. The slopes are converted to a fractional formation of $ClNO_2$ ($f$) via Eq 5.



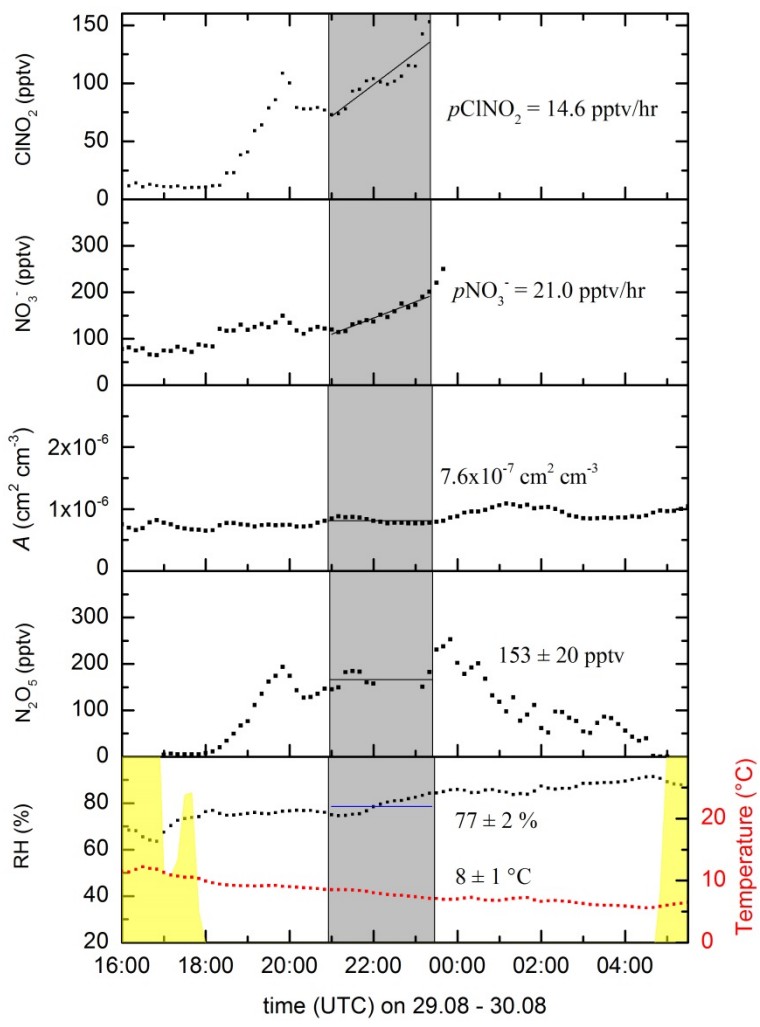

**Fig 6**. Plot demonstrating the derivation of $f$ and $\gamma$ from $ClNO_2$ and $NO_3^-$ measurements using analysis method 1b. The grey area represents the time period in which $pClNO_2$ and $pNO_3^-$ were measured. Average values of RH, $[N_2O_5]$ and $A$ over the same period are also indicated .


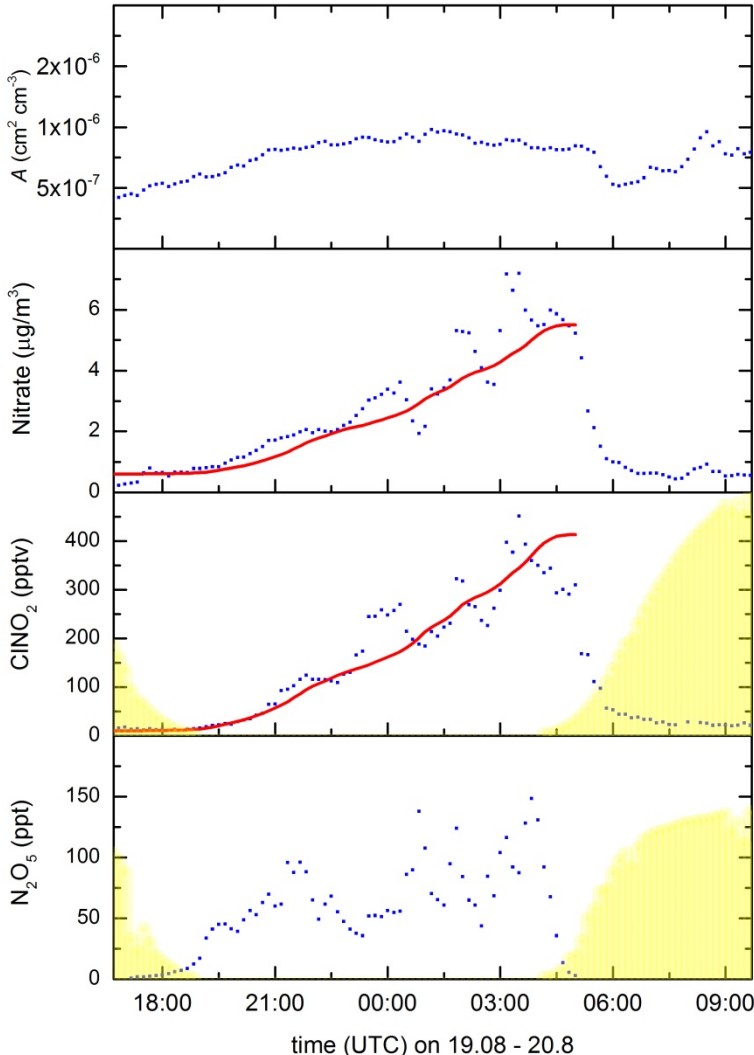

**Fig. 7**. Plot demonstrating the derivation of $f$ and $\gamma$ from $ClNO_2$ and $NO_3^-$ measurements using analysis method 1c. The red lines are the predicted, integrated concentrations of $NO_3^-$ and $ClNO_2$ calculated using the summed particle surface area ($A$) from both coarse and fine particles. In this particular case the analysis returns a value of $\gamma = 0.085$ and $f = 0.33$. The relative humidity over this period was $82 \pm 2$ %.





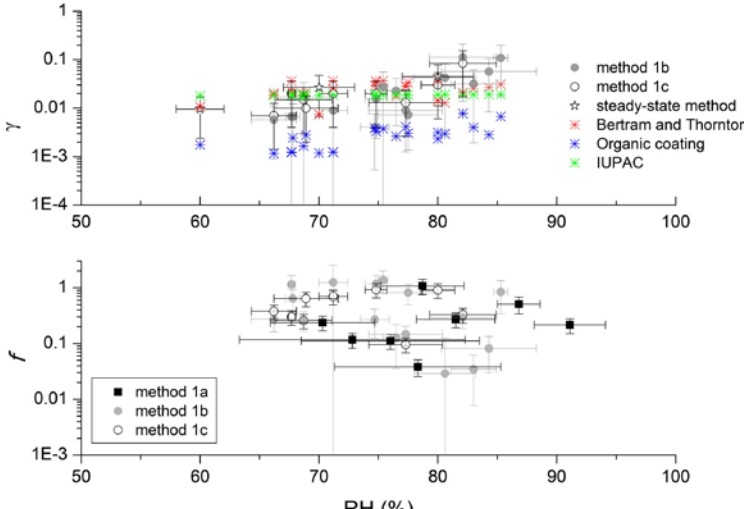

**Fig. 8**. Values of γ and *f* derived via methods 1a-1c and the steady-state analysis during PARADE. The vertical error bars represent total uncertainty including errors associated with the measurement of concentrations of trace gases and particle surface areas. The horizontal error bars are the standard deviation of the relative humidity over the measurement period (1-3 hours for methods 1b and 1c and up to 8 hours for method 1a.). The red stars uptake coefficients calculated using the Bertram and Thornton (2009) parametrisation, the blue stars include the impact of an organic coating (Anttila et al, 2006).



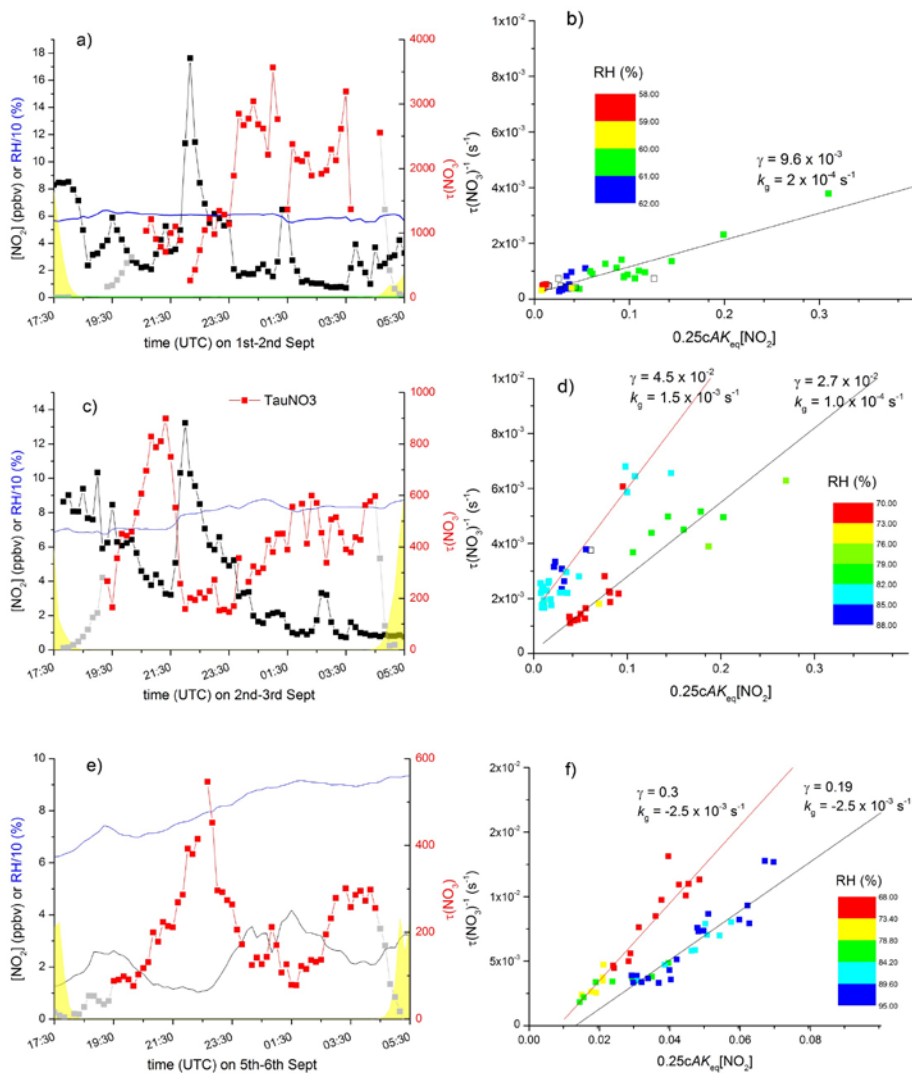

**Fig. 9**. Derivation of γ via analysis of NO$_3$ steady-state lifetime variation with NO$_2$. The grey τ(NO$_3$) data points were not considered in the analysis as NO$_3$ will not always be in steady state in the first hours after sunset or during sunrise.



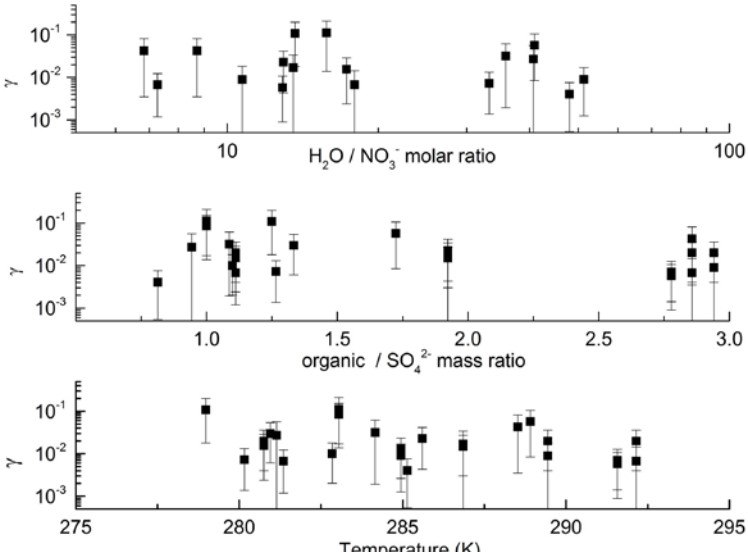

**Fig. 10**. Plot of $\gamma$ versus temperature, molar ratio of particle $H_2O$ to particulate nitrate and the particle organic to $SO_4^{2-}$ mass-ratio. Error bars are total uncertainty, calculated as described in the text.

