# Peer review of "Estimating N2O5 uptake coefficients using ambient measurements of NO3, N2O5, ClNO2 and particle-phase nitrate."

_Atmospheric Chemistry and Physics, 2016_

## Referee Comment (RC1) · Anonymous Referee #1 · 6 Sep 2016

General comments

This manuscript reports estimates of uptake coefficients of N2O5 and yields of ClNO2 based on measured data of the major nitrogen oxides species, NO3, N2O5 and ClNO2, as well as of important aerosol characteristics, including nitrate (but unfortunately not chloride) content, from Kleiner Feldberg observatory, Germany, during the 2011 PA-RADE campaign. Two methods were employed to derive heterogeneous reaction rates: the first one relied on the formation rates of ClNO2 and particulate nitrate, the second one on the steady state life time of NO3. The results for the uptake coefficients turn out to be in good agreement with uptake coefficients as derived from laboratory experiments and taking into account nitrate and estimated chloride contents and using

parameterizations either by the IUPAC evaluation or by Bertram and Thornton. The fractional ClNO2 yields are consistent with the highly variable (and usually low) particulate chloride contents at this site.

In spite of the fact that no surprises with respect to the multiphase chemistry of N2O5 are found, this work is a very valuable and timely contribution, since the heterogeneous fate of N2O5 and its impact on NOx, O3 and partially the halogen budgets is poorly constrained. While laboratory studies on this subject provide important cornerstones on kinetics and yields for model systems, the complexity of particulate matter and feedbacks with gas phase cycles, including uncertainties related to NO3 sinks in the gas phase, render it very important to analyze measured field data to test the hypotheses established based on laboratory studies. The manuscript is a very detailed account of a rigorous and careful analysis of the PARADE data that includes also thoughtful consideration of the uncertainties related to complex meteorology and air mass histories. The manuscript is overall well written and structured; in parts the reader may get lost in the details of the analysis and would appreciate some help to get back on the storyline. The specific comments below are of minor concern, but could help to polish and streamline the manuscript.

1) Figure 1, shown in the context of the introduction, could be more specific in the particulate phase chemical reactions, i.e., include R1-R3; maybe then also the parameter f could be indicated where water and chloride compete for nitronium ion. If the figure is in one-column width, this would still fit within the same size of particle.

2) page 8, lines 24 and below: the degassing of HNO3 following uptake of N2O5 and nitrate formation via uptake of HNO3 from the gas phase are strongly linked to the aerosol pH, which is influenced by both airmass history longer back as well as recent N2O5 uptake. If HNO3 in the gas phase is not partitioning to the aerosol phase, HNO3 deriving from N2O5 would also be likely be degassed. The rather lengthy discussion of the two assumptions continuing on the next page is not really coming back to this point.

3) page 12, around line 28: are VOC data available that could help explaining the variability of the NO3 lifetime? A note on this is mentioned at the end of section 4.3

4) section 4.4 appears partly repetitive, since to some degree most of the issues are already addressed in the detailed discussion of the different analysis methods. To this reviewer, this section is interrupting a bit the sequence of sections 4.1, 4.2., then continued in sections 4.5 and 4.6. The authors could consider integrating the additional aspects into the sections before or after to improve overall text flow.

5) page 14, line 30: this first summary statement could be expressed in a more positive mood. It is a substantial result that consistency with lab observation is achieved! So, the value of finding this consistence could be emphasized more.

6) page 15, line 5: low f values may also result from the presence organics nitrated by nitronium, in spite of the presence of chloride.

7) page 16, description of parameterization: it is not explicitly stated how [H2O] was derived for the IUPAC paremeterization; this is only mentioned for the Bertram and Thornton parameterization further below.

8) page 17 (and elsewhere), discussion of Figure 8: the data derived from this study show (to the eye) a trend of gamma increasing with humidity. Is this trend significant? The same trend is not apparent from the accepted paremeterizations. This could maybe be addressed in a short discussion.

9) page 17, line 23: Grzinic et al. (2015) should be added here in addition to mention that it is not just the oxidation state and reduced water concentration but also the higher viscosity (coming along with that) leading to lower uptake coefficients.

10) page 18, line 11 and below: As shown by Grzinic et al. (2015), the (volume) reaction limited regime may not be appropriate, since the increasing viscosity driven by oxidized organics leads rather to a decrease of the reacto-diffusive length.

11) Figure 9: would it be helpful to add aerosol surface to volume ratio to the plots to

make the correlation of the NO3 lifetime with A versus that with NO2 apparent in the data plots a), c) and e)? In plot e), add symbols for NO2 or remove them in the other plots as well to make consistent among all.

Technical comments

Page 9, line 10: . . .as N2O5 and HNO3 have completely different. . .

Page 12, line 13: . . .of this night and may also be. . .

Page 13, line 3: upper case K in Keq

Page 13, line 14: 'As the authors point out', maybe reiterate Wagner et al. (2013), since their mentioning is quite a few lines back

Page 14, line 21: . . .values reported here. . ., check: here or there (CalNex)

Page 15, line 20: sentence needs revision; . . . largely based on the relationship of ClNO2 with air mass origin?

Page 18, line 8: . . .setting f to one,. . .

References

Gržinić, G., Bartels-Rausch, T., Berkemeier, T., Türler, A., and Ammann, M.: Viscosity controls humidity dependence of N2O5 uptake to citric acid aerosol, Atmos. Chem. Phys., 15, 13615-13625, 2015

---

## Referee Comment (RC2) · R.A. Cox (Referee) · 7 Sep 2016

The paper describes a study involving field measurements of nitrogen oxides and atmospheric particulate matter, and analysis aimed at estimating the uptake coefficient (gamma) and yield of nitryl chloride (f) in the heterogeneous processing of dinitrogen pentoxide in the reaction: N2O5 + Cl- = ClNO2 + NO3-. The measurement site is at 800m altitude in a rural location in Western Germany, which is influenced by pollution from the adjacent Rhein-Main conurbation, and by long range transport of air of marine origin containing sea salt aerosol. This is the latest of several papers appearing in the literature reporting estimates of N2O5 uptake coefficients and reaction paths on ambient atmospheric aerosol of more or less defined composition. It is known from laboratory studies that the rates of these heterogenous reactions are highly dependent on atmospheric conditions as well as the chemical nature of the aerosol. Measurements under 'real world' conditions are necessary to provide confidence that the correct parameters are used. This is a timely study and the measurements have been conducted with a well conceived strategy for advancement of knowledge of this potentially important atmospheric process. Two different methods were used to obtain estimates of gamma and f from the observational data. In the first methodology (discussed in section 4.1) involved determination of the formation rates of ClNO2 and NO3- products for known [N2O5] and aerosol surface area. A total of 12 values of gamma were obtained during the field campaign. The second methodology used for estimating gamma is the so called steady state method, (discussed in section 4.2), which assumes that during nighttime the loss of NO3 and N2O5 is in balance with production from NO2 +O3. The gamma values can be evaluated from the expressions for inverse steady state lifetimes of NO3 and N2O5 Periods were selected so that air mass characteristics were consistent with assumptions made for a tractable analysis to obtain the target parameters. Only 3 values of gamma were obtained using the steady state method. As in previous studies in which uptake coefficient have been derived from ambient data sets there is large variability in the results over the time period of the observations: (0.004<gamma<0.11; gamma(av) =0.028 ± 0.027). Overall the results were consistent with those derived from earlier field studies, as discussed in section 4.4. The factors affecting the gamma values derived in this study were are discussed in section 4.6 in terms of parameterisations of IUPAC panel (RH), Bertram and Thornton (2009; particulate NO3-, Cl- and H2O), and Antilla (2006; organic content) derived from the body of information based on laboratory studies. The predictions and the measurements are again in reasonable agreement. However the apparent influence of particulate [NO3-] is less than predicted by the models and the predicted suppression of uptake by organics is much less if it is assumed that the organic content exists as a hydrophobic coating. There was also a high variability in the values of f (range 0.035 ± 0.027 to 1.38 ± 0.60). The large spread of values is expected from the variability in particulate Cl-
as a result of differing air mass origins (discussed in section 4.5). The observation of ClNO2 is an unambiguous indication that the particles at the measurement site contain Cl-, which will influence the overall uptake rate in the field situation. In summary the study provides further evidence for a role of active heterogeneous chemical processing of N2O5 in the lower atmosphere at nighttime, which is strongly influenced by the local physical conditions and chemical composition of the particles. The rates are as a result very variable but current mechanisms based on laboratory studies give a reasonable rationale of the parameters observed.

Other comments The presentation of the work is of a high standard of accuracy and completeness. For such complex data set and depth of analysis it is reasonably easy to read and the diagrams are informative and not excessively detailed. The conclusions are justified in the context of the body of knowledge on this topic.

Minor queries P 3, l. 20 insert 'a' after 'in' p 5, l. 1 please use metric units of length paragraph starting l.28 and Fig 2. Please reorder the 3 plots to be consistent with the order they are discussed in the text p 8, l. 1 Please indicate here that the covariance is illustrated in Fig 5 p 17, l. 14 please indicate the direction of the dependence of gamma suppression on RH; does it increase or decrease? Table 1, last row, col. 1 Were the observations made at 650 m or at 825 m, the altitude referred to on p3, l.27?

Please also note the supplement to this comment:
http://www.atmos-chem-phys-discuss.net/acp-2016-693/acp-2016-693-RC2-supplement.pdf

---

## Author Response (AR1)

We thank the review 1 for this positive appraisal of our manuscript. Each comment made (black) and our response (blue) is listed below. Red text indicates where changes to the manuscript have been made.

This manuscript reports estimates of uptake coefficients of N2O5 and yields of ClNO2 based on measured data of the major nitrogen oxides species, NO3, N2O5 and ClNO2, as well as of important aerosol characteristics, including nitrate (but unfortunately not chloride) content, from Kleiner Feldberg observatory, Germany, during the 2011 PARADE campaign. Two methods were employed to derive heterogeneous reaction rates: the first one relied on the formation rates of ClNO2 and particulate nitrate, the second one on the steady state life time of NO3. The results for the uptake coefficients turn out to be in good agreement with uptake coefficients as derived from laboratory experiments and taking into account nitrate and estimated chloride contents and using parameterizations either by the IUPAC evaluation or by Bertram and Thornton. The fractional ClNO2 yields are consistent with the highly variable (and usually low) particulate chloride contents at this site. In spite of the fact that no surprises with respect to the multiphase chemistry of N2O5 are found, this work is a very valuable and timely contribution, since the heterogeneous fate of N2O5 and its impact on NOx, O3 and partially the halogen budgets is poorly constrained. While laboratory studies on this subject provide important cornerstones on kinetics and yields for model systems, the complexity of particulate matter and feedbacks with gas phase cycles, including uncertainties related to NO3 sinks in the gas phase, render it very important to analyze measured field data to test the hypotheses established based on laboratory studies. The manuscript is a very detailed account of a rigorous and careful analysis of the PARADE data that includes also thoughtful consideration of the uncertainties related to complex meteorology and air mass histories. The manuscript is overall well written and structured; in parts the reader may get lost in the details of the analysis and would appreciate some help to get back on the storyline. The specific comments below are of minor concern, but could help to polish and streamline the manuscript.

We thank the reviewer for this very positive overall assessment of the paper.

1) Figure 1, shown in the context of the introduction, could be more specific in the particulate phase chemical reactions, i.e., include R1-R3; maybe then also the parameter f could be indicated where water and chloride compete for nitronium ion. If the figure is in one-column width, this would still fit within the same size of particle.

The suggested changes to Figure 1 have been made and the aqueous phase chemistry is now illustrated in more detail.

2) page 8, lines 24 and below: the degassing of HNO3 following uptake of N2O5 and nitrate formation via uptake of HNO3 from the gas phase are strongly linked to the aerosol pH, which is influenced by both airmass history longer back as well as recent N2O5 uptake. If HNO3 in the gas phase is not partitioning to the aerosol phase, HNO3 deriving from N2O5 would also be likely be degassed. The rather lengthy discussion of the two assumptions continuing on the next page is not really coming back to this point.

We have indicated how the degassing of particulate nitrate as HNO3 and the nighttime uptake of HNO3 would affect our results. Based on measurements during a subsequent campaign we find no strong indication that our calculations are strongly biased, but, in the absence of simultaneous gas-phase HNO3 measurements cannot prove this. The lengthy discussion is an honest airing of these issues which we prefer not to understate.

3) page 12, around line 28: are VOC data available that could help explaining the variability of the NO3 lifetime? A note on this is mentioned at the end of section 4.3

As is generally the case, the VOC measurements (GC) were generally made at low time resolution and do not shed light on this issue.

4) section 4.4 appears partly repetitive, since to some degree most of the issues are already addressed in the detailed discussion of the different analysis methods. To this reviewer, this section is interrupting a bit the sequence of sections 4.1, 4.2., then continued in sections 4.5 and 4.6. The authors could consider integrating the additional aspects into the sections before or after to improve overall text flow.

We agree and the sequence has been changed so that section  $4.4 \pmod{4.6}$  has been moved after sections  $4.5 \pmod{4.4}$  and section  $4.6 \pmod{4.5}$ .

5) page 14, line 30: this first summary statement could be expressed in a more positive mood. It is a substantial result that consistency with lab observation is achieved! So, the value of finding this consistence could be emphasized more.

This section and this statement has now been moved to the end of the document, just before the conclusions. Whilst we indicate that some ambient datasets confirm the laboratory observation of  $\gamma$  suppression by organics and/or nitrate, others do not. We also do not know what drives the high variability in the ambient measurements of gamma. Clearly, more ambient datasets are urgently required before we can conclude that we have reached a level of understanding that enables prediction of N2O5 uptake coefficients in different environments. We have added a sentence to the conclusions to indicate this "There is an urgent need for further laboratory work on synthetic aerosols and more field measurements that investigate the uptake of N2O5 in different "real-world" environment, especially chemically complex ones as found in the continental boundary layer."

6) page 15, line 5: low f values may also result from the presence of organics nitrated by nitronium, in spite of the presence of chloride.

Agreed. We now write: "We thus expect *f* to be largest in polluted coastal regions (unless there is a large organic content that can react with  $H_2NO_3^+$ ) and lowest (or zero) in continental regions with no marine influence or anthropogenic chlorine emissions."

7) page 16, description of parameterization: it is not explicitly stated how [H2O] was derived for the IUPAC parameterization; this is only mentioned for the Bertram and Thornton parameterization further below.

We now mention how liquid water constant was also calculated for the IUPAC parameterisation.: "IUPAC preferred values are listed (for ammonium sulphate) as  $\alpha_b = 0.03$ ,  $k_{H2O} = 1.0 \times 10^5 \text{ M}^{-1} \text{s}^{-1}$ ,  $D_1 = 1 \times 10^{-5} \text{ cm}^2 \text{ s}^{-1}$  and  $\text{H} = 2 \text{ M} \text{ atm}^{-1}$ . Particle liquid water content [H2O(*l*)], was calculated with the E-AIM model (http://www.aim.env.uea.ac.uk/aim/model3/model3a.php) (Clegg et al., 1998; Wexler and Clegg, 2002) using particulate nitrate, sulphate and ammonium concentrations measured by the AMS and the relative humidity." 8) page 17 (and elsewhere), discussion of Figure 8: the data derived from this study show (to the eye) a trend of gamma increasing with humidity. Is this trend significant? The same trend is not apparent from the accepted parameterizations. This could maybe be addressed in a short discussion.

We have added text to highlight this point and provide a potential explanation:

"..... the BT- parameterisation lies at the upper range of the measurements for relative humidities between 65 and 75 % and at the lower end for the highest relative humidities encountered, which may indicate a positive dependence of measured  $\gamma$  on RH which is not predicted by the parameterisations. As the particles during PARADE have significant organic content (see Fig. 2) this may reflect the fact that the organic suppression of  $\gamma$  is reduced at high relative humidity as reported by (Gaston et al., 2014). We note however that the values of  $\gamma$  measured at large RH are larger than most measurements on pure laboratory samples, which may indicate a measurement bias under some conditions.

9) page 17, line 23: Grzinic et al. (2015) should be added here in addition to mention that it is not just the oxidation state and reduced water concentration but also the higher viscosity (coming along with that) leading to lower uptake coefficients.

We now write: "As discussed by Gaston et al. (2014) and Grzinic et al. (2015), the effect of organic content in a particle will depend on the amount and oxidation state of the organics, with both solubility and viscosity impacting on the response of  $\gamma$  to changes in relative humidity."

10) page 18, line 11 and below: As shown by Grzinic et al. (2015), the (volume) reaction limited regime may not be appropriate, since the increasing viscosity driven by oxidized organics leads rather to a decrease of the reacto-diffusive length.

We have now added the sentence: as discussed by Grzinic et al. (2015), the increasing viscosity driven by the presence of oxidized organics may lead to a reduction in diffusive transport into the particle.

11) Figure 9: would it be helpful to add aerosol surface to volume ratio to the plots to make the correlation of the NO3 lifetime with A versus that with NO2 apparent in the data plots a), c) and e)?

In plot e), add symbols for NO2 or remove them in the other plots as well to make consistent among all.

The aerosol surface area does not change much over these periods. To preserve clarity of presentation, we prefer not to add more detail to these plots.

We have made the representation of the NO2 data consistent (with symbols).

Technical comments

Page 9, line 10: as N2O5 and HNO3 have completely different

Correction made

Page 12, line 13: of this night and may also be

Correction made

Page 13, line 3: upper case K in Keq

Correction made

Page 13, line 14: 'As the authors point out', maybe reiterate Wagner et al. (2013), since their mentioning is quite a few lines back

We now write: "as Wagner at al. (2013) point out"

Page 14, line 21: values reported here, check: here or there (CalNex)

To remove ambiguity we now write: values reported in the present work were derived...

Page 15, line 20: sentence needs revision; largely based on the relationship of ClNO2 with air mass origin?

We now write: "which is largely based on the dependence of  $CINO_2$  on air mass origin..." Page 18, line 8: setting f to one

Correction made

We thank Tony Cox for this positive appraisal of our manuscript. Each comment made (black) and our response (blue) is listed below. Red text indicates where changes to the manuscript have been made.

The paper describes a study involving field measurements of nitrogen oxides and atmospheric particulate matter, and analysis aimed at estimating the uptake coefficient (gamma) and yield of nitryl chloride (f) in the heterogeneous processing of dinitrogen pentoxide in the reaction: N2O5 + Cl = ClNO2 + NO3. The measurement site is at 800m altitude in a rural location in Western Germany, which is influenced by pollution from the adjacent Rhein-Main conurbation, and by long range transport of air of marine origin containing sea salt aerosol. This is the latest of several papers appearing in the literature reporting estimates of N2O5 uptake coefficients and reaction paths on ambient atmospheric aerosol of more or less defined composition. It is known from laboratory studies that the rates of these heterogeneous reactions are highly dependent on atmospheric conditions as well as the chemical nature of the aerosol. Measurements under 'real world' conditions are necessary to provide confidence that the correct parameters are used. This is a timely study and the measurements have been conducted with a well conceived strategy for advancement of knowledge of this potentially important atmospheric process. Two different methods were used to obtain estimates of gamma and f from the observational data. In the first methodology (discussed in section 4.1) involved determination of the formation rates of CINO2 and NO3- products for known [N2O5] and aerosol surface area. A total of 12 values of gamma were obtained during the field campaign. The second methodology used for estimating  $\gamma$  is the so called steady state method, (discussed in section 4.2), which assumes that during nighttime the loss of NO3 and N2O5 is in balance with production from NO2 +O3. The gamma values can be evaluated from the expressions for inverse steady state lifetimes of NO3 and N2O5 Periods were selected so that air mass characteristics were consistent with assumptions made for a tractable analysis to obtain the target parameters. Only 3 values of gamma were obtained using the steady state method. As in previous studies in which uptake coefficient have been derived from ambient data sets there is large variability in the results over the time period of the observations:  $(0.004 < \text{gamma} < 0.11; \text{gamma}(\text{av}) = 0.028 \pm 0.027)$ . Overall the results were consistent with those derived from earlier field studies, as discussed in section 4.4. The factors affecting the gamma values derived in this study are discussed in section 4.6 in terms of parameterisations of IUPAC panel (RH), Bertram and Thornton (2009; particulate NO3-, Cl- and H2O), and Antilla (2006; organic content) derived from the body of information based on laboratory studies. The predictions and the measurements are again in reasonable agreement. However the apparent influence of particulate [NO3-] is less than predicted by the models and the predicted suppression of uptake by organics is much less if it is assumed that the organic content exists as a hydrophobic coating. There was also a high variability in the values of f (range 0.035  $\pm$  0.027 to 1.38  $\pm$  0.60). The large spread of values is expected from the variability in particulate Cl- as a result of differing air mass origins (discussed in section 4.5). The observation of ClNO2 is an unambiguous indication that the particles at the measurement site contain Cl-, which will influence the overall uptake rate in the field situation. In summary the study provides further evidence for a role of active heterogeneous chemical processing of N2O5 in the lower atmosphere at nighttime, which is strongly influenced by the local physical conditions and chemical composition of the particles. The rates are as a result very variable but current mechanisms based on laboratory studies give a reasonable rationale of the parameters observed

Other comments. The presentation of the work is of a high standard of accuracy and completeness. For such complex data set and depth of analysis it is reasonably easy to read

and the diagrams are informative and not excessively detailed. The conclusions are justified in the context of the body of knowledge on this topic

We thank Tony Cox for this very positive overall assessment of the paper.

Minor queries

P 3, l. 20 insert 'a' after 'in'

Correction made

p 5, l. 1 please use metric units of length paragraph starting 1.28

We have added mm lengths to the inch lengths listed.

Fig 2. Please reorder the 3 plots to be consistent with the order they are discussed in the text We prefer to keep the present order.

p 8, l. 1 Please indicate here that the covariance is illustrated in Fig 5

Fig 5 shows the correlation. The co-variance is apparent from the time series.

p 17, l. 14 please indicate the direction of the dependence of gamma suppression on RH; does it increase or decrease?

We now write "As the particles during PARADE have significant organic content (see Fig. 2) this may reflect the fact that the organic suppression of  $\gamma$  is reduced at high relative humidity as reported by (Gaston et al., 2014)."

Table 1, last row, col. 1 Were the observations made at 650 m or at 825 m, the altitude referred to on p3, 1.27?

The observations were at 825 m. This was a typographical error and has been amended.

**Estimating N2O5 uptake coefficients using ambient measurements of NO3, N2O5, ClNO2 and particle-phase nitrate.**

G. J. Phillips1, J. Thieser1, M. J. Tang1, N. Sobanski1, G. Schuster1, J. Fachinger2, F. Drewnick2, S. Borrmann2, H. Bingemer3, J. Lelieveld1, and J. N. Crowley1

[revised manuscript text omitted]

$$NO_2 + O_3 \longrightarrow NO_3 + O_2$$
 (R7)

$$NO_2 + NO_3 + M \longrightarrow N_2O_5 + M$$
(R8)

$$N_2O_5 + M \longrightarrow NO_2 + NO_3 + M$$
 (R9)

30 Ambient concentrations of NO3 and  $N_2O_5$  are thus coupled via the gas-phase, thermochemical equilibrium that exists due to R8 and R9, so that the relative amounts of NO3 and  $N_2O_5$  are determined by temperature and NO2 levels.

The so called "steady state" determination method for  $\gamma$  is based on the assumption that, after a certain period of time following sunset (often on the order of hours) the direct and indirect losses of NO3 and N2O5 balance their production. The steady-state lifetimes can then be calculated from observations of the NO3 and N2O5 concentrations and the production term  $k_7[NO_2][O_3]$ , where  $k_7$  is the rate constant for reaction R7. This method was first used by Platt and co-workers to assess the

- 5 heterogeneous loss of N2O5 (Platt and Heintz, 1994; Platt and Janssen, 1995; Heintz et al., 1996), and has been extended by Brown and co-workers to derive γ in regions distant from NOx sources such as the marine environment (Aldener et al., 2006) and aloft (Brown et al., 2006; Brown et al., 2009) and most recently for the continental boundary layer (Brown et al., 2016a). [NO2] and [O3] measurements are required to calculate the rate of NO3 production, which is generally assumed to be via R7 only, though a contribution of NO2 oxidation by stabilised Criegee intermediates has recently been hypothesised to represent
- 10 a potential bias to this calculation (Sobanski et al., 2016). The steady-state analysis does not require any information about products formed by  $N_2O_5$  heterogeneous reactions.

The inverse steady-state lifetimes of NO3 ( $\tau_{NO3}$ ) and N2O5 ( $\tau_{N_2O_5}$ ) are given by expressions Eq.7 and Eq.8, respectively:

$$(\tau_{NO3})^{-1} \approx \gamma \left( 0.25\bar{c}AK_{eq}[NO_2] \right) + k_g \tag{7}$$

$$\left(\tau_{N_2 O_5}\right)^{-1} \approx k_g \left(K_{eq}[N O_2]\right)^{-1} + 0.25\bar{c}A\gamma$$
(8)

- 15 Where  $K_{eq}$  is the temperature dependent equilibrium constant describing the relative concentrations of NO2, NO3 and N2O5 (R8 and R9), [NO2] is the concentration of NO2, and  $k_g$  is the pseudo first-order loss constant for NO3 loss in gas-phase reactions (e.g. with NO or with hydrocarbons). A plot of  $(\tau_{NO3})^{-1}$  or  $(\tau_{N_2O_5})^{-1}K_{eq}[NO_2]$  against  $0.25\bar{c}AK_{eq}[NO_2]$  should be a straight line with  $\gamma$  as slope and  $k_g$  as intercept. This method thus relies on the fact that the relative concentrations of N2O5 and NO3 vary with  $K_{eq}[NO_2]$  and thus the contribution of their individual losses to the overall lifetime of both NO3 and
- 20 N2O5 also varies with [NO2] once changes in temperature (and thus  $K_{eq}$ ) are accounted for. The method therefore assumes that, for a given analysis period in which NO2 is changing sufficiently to change the relative loss rates of NO3 and N2O5, both  $\gamma$  and  $k_g$  are constant (i.e. do not depend on [NO2]). This will not always be the case and we have often observed that the relation between  $(\tau_{NO3})^{-1}$  and  $K_{eq}$ [NO2] is non-linear. In environments where NO3 losses are dominated by gas-phase reactions of NO3, the uncertainty associated with the derivation of  $\gamma$  via a steady-state analysis is clearly greatly enhanced.
- Figure 9 illustrates some of these issues for data obtained over a period of several hours on the nights 1-2.09, 2-3.09 and 5-6.09. The first two nights (1-2.09 and 2-3.09) had long NO3 lifetimes and were selected for analysis as long NO3 lifetimes represent periods with low rates of NO3 loss by gas-phase processes (i.e.  $k_g$  is small). We have previously shown (Sobanski et al., 2016) that the long NO3-lifetimes on these nights result from sampling from a low altitude residual layer. A large variability in the nighttime NO2 mixing ratio on these nights (~1 to 13 ppbv, see Fig. 9a and 9c) should result in a significant
- 30 shift in the NO2-to-N2O5 ratio and thus sensitivity in the NO3 lifetime to the uptake of N2O5 to aerosol. On the night 1-2nd Sept., plume like features in the NO2 mixing ratio at ~22:00, 01:30 and 04:00 were accompanied by decreases in the steadystate NO3 lifetime. The inverse NO3 lifetime is plotted against  $0.25\bar{c}AK_{eq}[NO_2]$  in Fig. 9b. Here we have selected data that

was obtained from about 2 hr after sunset to the next dawn when NO3 started to decrease as represented by the red data points in Fig. 9a. The slope of the plot results in values of  $\gamma = (9.6 \pm 0.7) \times 10^{-3}$  and  $k_g = (2.0 \pm 0.6) \times 10^{-4} \text{ s}^{-1}$ , where the errors are statistical only. Over the period analysed, the relative humidity (blue line in Fig. 9a) was  $60 \pm 2\%$ .

On the night 2-3.09 at about 19:30 UTC ( $\approx$  1 hour after sunset), the NO3 lifetime increases gradually to a value of ~ 900 s

- 5 until 21:30 as NO2 decreases from ~ 7 to 3 ppbv. A rapid increase in NO2 at 21:30 is then accompanied by a much shorter NO3 lifetime. After ~22:00, NO2 slowly decreases and the NO3 lifetime recovers to about 500 s. Thus, also in this night there is a clear anti-correlation between NO2 and the NO3 lifetime. Figure 9d plots the inverse of the NO3 lifetime versus  $0.25\bar{c}AK_{eq}[NO_2]$  with the data point colour coded according to relative humidity. The first period of the night (orange, yellow and green data-points) are best described (black line) by an uptake coefficient of  $(2.7 \pm 0.2) \times 10^{-2}$  and  $k_g = (1.0 \pm 10^{-2})$
- 10 0.4) × 10-4 s-1. The second period (starting ~ 5 hours after sunset) is best characterised by larger values of  $\gamma = (4.5 \pm 0.3) \times 10^{-2}$  and  $k_g = (1.5 \pm 0.1) \times 10^{-3}$  s-1 (errors are statistical only). We note that prior to the peak in NO2 at 21:30 the relative humidity of the air was stable at  $\approx 70 \pm 3$  % whereas after 22:00 it remained at 80  $\pm 3$  %. A shift in air mass to one with larger water vapour content could help explain the larger values of  $\gamma$  obtained in the 2nd period of this night and may also be the reason for the larger gas-phase losses of NO3 if the more humidified air mass is more impacted by boundary-layer emissions. On two other nights when the NO3 lifetimes were long (30-31.08 and 31.08-1.09) there was very little variation in

NO2 so that the steady-state lifetime of NO3 was insensitive to the uptake coefficient. In Fig. 9e and 9f we present data that were obtained with sufficient variation in NO2, but with relatively short NO3 lifetimes. As in the datasets discussed above, NO3 lifetimes are anti-correlated with NO2. 
[revised manuscript text omitted]
                     | γ                               | Notes                                                                                              | reference               |
|------------------------------|---------------------------------|----------------------------------------------------------------------------------------------------|-------------------------|
| (location / height)          | (method)                        |                                                                                                    |                         |
| Ship                         | $0.03 \pm 0.02$                 | Polluted marine                                                                                    | (Aldener et al., 2006)  |
| (East Coast U.S., 11 m)      | (ss)                            |                                                                                                    |                         |
| Aircraft                     | < 0.0016 - 0.017                | Continental / coastal residual                                                                     | (Brown et al., 2006)    |
| (NE-U.S. < 1.5 km)           | (\$\$)                          | layer. $\gamma$ dependence on organic / sulphate content.                                          |                         |
| Aircraft                     | 0.0005-0.006                    | Continental / coastal residual                                                                     | (Brown et al., 2009)    |
| (Texas, U.S. < 1 km)         | (ss)                            | layer. $\gamma$ variable but independent of RH or aerosol composition                              |                         |
| Ground                       | < 0.01-0.04                     | Urban / sub-urban environment. $\gamma$                                                            | (Bertram et al., 2009b) |
| (US, Seattle/Boulder 5-10 m) | (AFR)                           | dependence on organic / sulphate ratio and RH.                                                     |                         |
| Ground                       | < 0.001-0.029                   | Polluted coastal environment. $\gamma$                                                             | (Riedel et al., 2012b)  |
| (US, La Jolla, 15 m)         | (AFR)                           | dependence on nitrate content                                                                      |                         |
| Ground                       | 0.002-0.1                       | Continental, pollution impacted                                                                    | (Wagner et al., 2013)   |
| (U.S, Boulder 10-300 m)      | (Box model)                     | boundary layer / residual layer. $\gamma$ dependence on nitrate content                            |                         |
| Aircraft                     | 0.01-0.03                       | Continental, pollution impacted                                                                    | (Morgan et al., 2015)   |
| (N.W. Europe 500-1000 m)     | (ss)                            | residual layer / free troposphere. $\gamma$ dependence on nitrate content                          |                         |
| Ground                       | $\gamma f = 0.008 \text{ (av)}$ | Coastal (CalNex-LA). $\gamma f$ was                                                                | (Mielke et al., 2013)   |
| US, Pasadena (10 m)          |                                 | suppressed by particle organic content and enhanced by                                             |                         |
|                              |                                 | particulate chloride content.                                                                      |                         |
| Ground                       | 0.004–0.029                     | Coastal, pollution impacted mountain site.                                                         | (Brown et al., 2016a)   |
| China, Hong Kong, (957 m)    | (ss)                            |                                                                                                    |                         |
| Ground                       | 0.004–0.11                      | Semi-rural mountain site with
anthropogenic influence, mixed
boundary layer / residual layer | This work               |
| (S.W. Germany, 825 m)        | (1b, 1c, ss)                    |                                                                                                    |                         |

**Notes**: ss = steady-state analysis. AFR = aerosol flow reactor. Av = averaged over a campaign.

Fig. 1. Gas- and aqueous-phase chemical processes forming  $CINO_2$  and particulate nitrate indicating the role of  $CINO_2$  in modifying the effect of